# How language models extrapolate outside the training data: A case study in Textualized Gridworld

## Abstract

Language models' ability to extrapolate learned behaviors to novel, more complex environments beyond their training scope is highly unknown. This study introduces a path planning task in a textualized Gridworld to probe language models' extrapolation capabilities. We show that conventional approaches, including next-token prediction and Chain of Thought (CoT) fine-tuning, fail to extrapolate in larger, unseen environments. Inspired by human cognition and dual-process theory, we propose *cognitive maps for path planning*—a novel CoT framework that simulates human-like mental representations. Our experiments show that cognitive maps not only enhance extrapolation to unseen environments but also exhibit human-like characteristics through structured mental simulation and rapid adaptation. Our finding that these cognitive maps require specialized training schemes and cannot be induced through simple prompting opens up important questions about developing general-purpose cognitive maps in language models. Our comparison with exploration-based methods further illuminates the complementary strengths of offline planning and online exploration.

## 1 Introduction

### 1.1 Difference between human cognition and language models: cognitive map

Recent advancements in language models have demonstrated remarkable proficiency across various complex tasks, ranging from natural language understanding to code generation (Brown et al., 2020; Touvron et al., 2023; Chowdhery et al., 2023; Chen et al., 2021). These models, primarily trained on next-token prediction, excel in general planning tasks by leveraging their extensive learned knowledge and pattern recognition abilities (Ahn et al., 2022; Liang et al., 2023; Song et al., 2023). However, they often struggle with robust, long-horizon planning tasks. Experiments in controlled settings, such as Einstein's puzzle, digit multiplication (Dziri et al., 2024), or Blocksworld (Valmeekam et al., 2022), demonstrate that even advanced models like GPT-4 (OpenAI et al., 2024) fail to generalize effectively to compositional tasks when faced with longer, more complex planning scenarios. In contrast, humans excel in such tasks, solving them effortlessly even in challenging environments.

What cognitive mechanisms enable humans to generalize so effectively? According to the dual-process theory, humans employ two types of reasoning: fast, automatic System 1 processes and slower, deliberative System 2 processes (Kahneman, 2011). System 2 reasoning, characterized by constructing and utilizing mental representations, allows humans to adapt learned behaviors to novel, complex environments (Yousefzadeh & Mollick, 2021; Fellini & Morellini, 2011; Stojic et al., 2018). Cognitive science literature refers to this aspect of human cognition as *cognitive maps*—mental representations of environments that enable flexible planning. These cognitive maps involve a network of brain regions, including the hippocampus, prefrontal cortex, and parietal cortex, and play a critical role in complex decision-making and adaptation to new situations (O'Keefe & Nadel, 1978; Daw et al., 2005; Doll et al., 2012).

While the next-token prediction paradigm equips language models to perform System 1-like pattern recognition, it does not naturally encompass System 2 processes. Recent approaches, such as Chain of Thought (CoT) reasoning, show promise for mimicking System 2 reasoning via in-context

learning (Wei et al., 2023), prompting (Kojima et al., 2023), fine-tuning (Kim et al., 2023), or even without explicit prompting (Wang & Zhou, 2024). However, conventional CoT methods fall short of exhibiting cognitive map-like behavior (Momennejad et al., 2023). In essence, while these approaches improve reasoning capabilities, they still fall short of implementing true cognitive maps that enable flexible, map-like planning and navigation of problem spaces.

## 1.2 Testing cognitive maps for language models through extrapolability

Before asking how to create cognitive maps for language models, we first need to establish a method to test whether a language model possesses a cognitive map. We propose that extrapolability—the ability to generalize learned knowledge to novel, complex environments after training on simple demonstrations—can serve as a proxy for this[1].

We posit that the fundamental distinction between AI models and human cognition lies in extrapolation rather than interpolation capabilities. While language models excel at interpolation—performing effectively on tasks within the training distribution due to KL divergence loss minimization during training (LeCun et al., 2015)—they struggle with extrapolation, particularly in compositional tasks or novel, complex environments.

To investigate cognitive maps in language models, we evaluated their extrapolation capabilities using a textualized Gridworld path-planning task (Brown, 2015). This choice aligns with cognitive science foundations, where spatial tasks have proven valuable for studying mental representations (Epstein et al., 2017; O'Keefe & Nadel, 1978; Kessler et al., 2024; Kadner et al., 2023). We designed controlled environments to examine models' ability to construct and utilize cognitive maps rather than focusing on scalability. Our experiments reveal that language models, whether trained via imitation learning or conventional Chain of Thought fine-tuning, fail to effectively extrapolate to larger environments (Figure 1).

## 1.3 Cognitive maps for language models in Textualized Gridworld setting

Then is it possible to inject human-like cognitive maps into language models? Theoretical findings suggest that complex problems outside the $TC^0$ (polynomial-sized constant-depth circuits) complexity class can be solved with an ample amount of CoT tokens (Merrill & Sabharwal, 2024; Feng et al., 2024). However, the precise *form* of CoT required remains unclear. Recent advancements, such as the GPT-o1 model (OpenAI, 2024), achieve remarkable performance across various domains, including mathematics and coding. Yet, its CoT is opaque, and the training data remains inaccessible, making it challenging to determine whether the model genuinely exhibits extrapolation.

Building upon prior research discussed in Appendix A, this paper presents a novel training approach for language models using datasets augmented with *cognitive maps for path planning* - as a form of Chain of Thought (CoT) reasoning. We design the cognitive maps to emulate human mental models used in spatial reasoning and decision-making by explicitly verbalizing complete decision trees (detailed in Section 3). Our experimental results demonstrate that fine-tuning models to generate such cognitive maps significantly improves their planning capabilities in novel, extrapolated environments. Importantly, unlike traditional CoT approaches that can be implemented through prompting alone, cognitive maps require specific training mechanisms. This finding addresses a critical gap identified in recent studies examining spatial reasoning and cognitive mapping capabilities in language models (Xu et al., 2024; Momennejad et al., 2023; Yamada et al., 2024; Li et al., 2024).

Our findings demonstrate that integrating cognitive maps into language models significantly enhances their capacity to generalize to complex and novel environments, bridging the gap between System 2 reasoning and extrapolability. Beyond emergent extrapolability, our cognitive maps exhibit

---

[1]It is worth noting that the concept of 'interpolation' and "extrapolation" can carry different meanings across AI research contexts, such as estimating values between known observations versus predicting future values in time series (Kolmogoroff, 1941; Wiener, 1949), or distinguishing between samples within versus outside the convex hull of the training data (Belkin et al., 2018; Bietti & Mairal, 2019; Adlam & Pennington, 2020; Balestriero et al., 2021). In this paper, we define these terms concerning training and application environments: "interpolation" refers to performance on environments similar to the training distribution, while "extrapolation" involves performance in novel, significantly different environments. Section 2.2 offers further discussion specific to this paper.

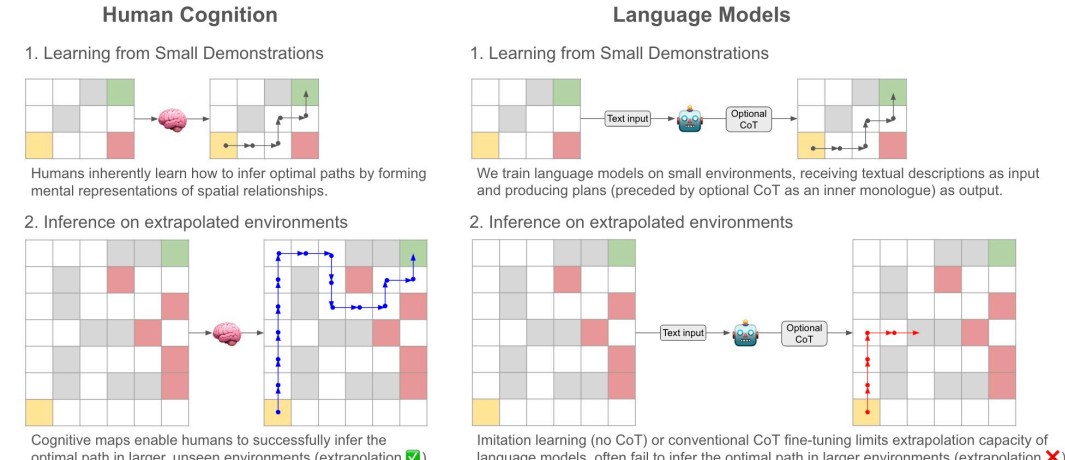

Figure 1: Comparing human cognition and language models for extrapolated path planning: Humans demonstrate robust extrapolation abilities in unseen grid environments due to cognitive maps that generalize spatial reasoning. In contrast, current language models struggle with extrapolation, highlighting the need for improved design of internal "thought" representations to achieve robust planning in novel scenarios rather than implicit or conventional CoT fine-tuning. **In this paper, we introduce *cognitive maps for path planning*, a specific form of inner monologues as a CoT that makes extrapolation happen for language models.** See Figure 2 and Section 3 for details.

key characteristics of human cognition: structured mental simulation during planning and rapid adaptation during training. Through comparison with exploration-based methods, we identify important trade-offs between planning efficiency and success rates, suggesting promising directions for hybrid approaches. Although this work is situated in the Gridworld domain, we believe these insights—particularly the unpromptable nature of cognitive maps and the need for specialized training schemes—provide valuable guidance for developing language models with more general-purpose cognitive architectures capable of human-like reasoning across diverse domains.

## 2 PROBLEM FORMULATION: PATH PLANNING IN TEXTUALIZED GRIDWORLD

### 2.1 DESCRIPTION OF THE TEXTUALIZED GRIDWORLD

We introduce Textualized Gridworld, inspired by Brown (2015), as our primary task. In this path planning challenge, an agent navigates from a start state to a goal state, avoiding obstacles. Each environment is designed with exactly one valid path to ensure clear evaluation of the model's planning capabilities.

The uniqueness of our approach lies in the textualized nature of both input and output. We provide the model with a textual description of the Gridworld environment, including the dimensions of the grid, the locations of the start and goal states, and the positions of obstacles. The model interacts with this environment by generating textual commands (either "up", "down", "left", or "right"), and receives textual descriptions of the resulting state transitions (See Appendix B.1 for examples of textualized input instructions and state descriptions).

Our primary objective is to investigate whether a language model trained on grid up to $n \times n$, can successfully plan paths in $a \times b$ grids, where $a$ and $b$ can vary in relation to $n$. We define interpolation when both $a$ and $b$ are smaller than or equal to $n$, and extrapolation when either $a$ or $b$ (or both) are greater than $n$. This challenge tests the model's ability to generalize its understanding of spatial relationships and apply path planning strategies beyond the scope of its training data.

Specifically, we train the model on 50,000 samples with grid sizes up to $10 \times 10$ for one epoch and tested on 3,000 samples with grid sizes up to $20 \times 20$, allowing us to evaluate the model's extrapolation capabilities. Also, we design each environment to have exactly one valid path to

ensure a clear evaluation of the model's planning ability. We describe further details such as setup information and data statistics in Appendices B.2 and B.4.

By framing this task in natural language, we explore the limits of language models' abstract thinking and problem-solving capacities, probing their ability to form and manipulate mental representations of complex spatial environments described solely through text.

## 2.2 JUSTIFICATION FOR CHOOSING TEXTUALIZED GRIDWORLD

The selection of Textualized Gridworld as our primary task is motivated by four key factors:

- **Cognitive science foundations:** Gridworld serves as an ideal testbed because probing mental representations through spatial tasks is foundational in cognitive science (Epstein et al., 2017; O'Keefe & Nadel, 1978; Kessler et al., 2024; Kadner et al., 2023). Similar to seminal studies in this field, our work leverages controlled environments to provide valuable insights into specific cognitive capabilities, such as the ability to construct and utilize cognitive maps. Through testing different CoT patterns during navigation tasks, we aim to identify representations that enable robust extrapolation capabilities.

- **Minimal knowledge requirements:** Gridworld provides an ideal environment to probe the extrapolability of language models while minimizing the influence of world knowledge. Unlike more complex board games such as Einstein's puzzle (Brainzilla, 2017) or Blocksworld (Valmeekam et al., 2022), Gridworld requires only a basic understanding of four actions (up, down, left, right) and simple, explicit state transitions. This simplicity allows us to focus on the model's ability to reason and plan, rather than its capacity to leverage pre-existing knowledge.

- **Scalability and generalization testing:** A crucial advantage of Gridworld is its ability to generate environments of arbitrary size. This feature is essential for testing the model's capacity to extrapolate beyond its training experience. Unlike other planning tasks such as coding or mathematical problem-solving, Gridworld allows us to systematically increase the problem space by expanding the grid dimensions. This scalability facilitates a clear and controlled assessment of the model's ability to generalize its planning strategies to larger, unseen environments.

- **Complexity class and computational limitations:** Gridworld's placement outside the $TC^0$ complexity class is crucial for our study. Unlike tasks such as multiplication or addition which fall within $TC^0$ and can be solved by embedding modification (McLeish et al., 2024) or CoT distillation (Deng et al., 2023; 2024), Gridworld requires more sophisticated computational processes. This characteristic makes it impossible to solve Gridworld through mere next-token prediction, as suggested by existing research on the limitations of fixed-precision transformer architectures (Merrill & Sabharwal, 2023). By choosing a task outside $TC^0$, we create a rigorous test of language models' capacity for complex reasoning and planning, challenging them to go beyond simple pattern recognition and construct multi-step plans. This allows us to explore the boundaries of what these models can achieve with their current architectures and training paradigms, potentially revealing insights into their ability to tackle more complex, sequential decision-making tasks.

Especially, we show that Gridworld environment complexity increases with grid size. We demonstrate systematic complexity differences between training and test datasets, highlighting the additional challenges posed by more complex Gridworld environments during inference. This analysis establishes that testing beyond the training bounds in Gridworld represents extrapolation not only in spatial scale but also in computational complexity. We provide a formal definition of environment complexity and detailed analysis in Appendix B.3.

## 3 COGNITIVE MAPS FOR PATH PLANNING

We now propose a universal design for cognitive maps for path planning as a CoT for language models. Our approach consists of three stages: sampling, propagation, and backtracking, enabling the model to construct a comprehensive mental representation and generate optimal solutions without external interaction.

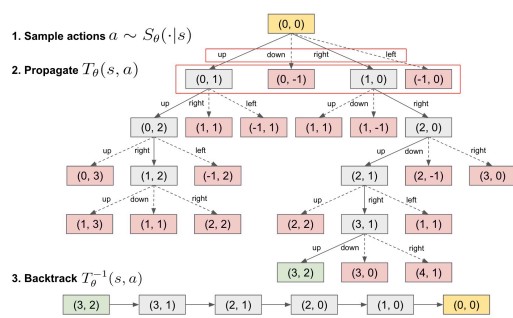 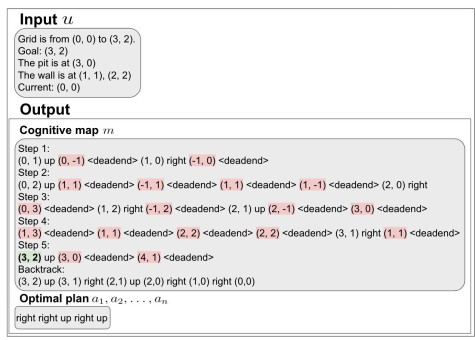

(a) Our design of cognitive map for path planning consists of three key steps: Sampling, Propagation, and Backtracking

(b) An example data instance for optimal planning, consists of input specification of the environment and the output consisting of CoT and optimal plan.

Figure 2: Cognitive maps for path planning: Our approach implements a tree-structured cognitive map that enables systematic exploration and path planning in Gridworld environments (a). The decision tree with the backtrack path is flattened into a Chain of Thought (CoT) format that language models can process as an inner monologue, capturing both the exploration process and path planning strategy (b). We show that this structured representation aims to enhance models' ability to reason about and solve navigation tasks even in larger, previously unseen environments.

1. **Sampling**: The model identifies potential actions for each state, sampling from $S(s) \subset A$ for the current state $s$. This process continues until the goal state is reached, encompassing a wide range of possible states within the world model.

2. **Propagation**: For each sampled action $a$ from state $s$, the model predicts the resulting state $T(s, a) \in \mathcal{S}$. In Gridworld, this involves simulating movements in four directions, considering obstacles and boundaries. This stage builds a global representation of the world model.

3. **Backtracking**: Once the goal is reached, the model retraces steps from the goal to the initial state, identifying the inverse transition $T^{-1}(s, a) \in \mathcal{S}$ for each state-action pair. This ensures path validity and optimality.

Constructing the cognitive map $m$ involves the sequential application of sampling ($S$), propagation ($T$), and backtracking ($T^{-1}$). We train the language model $\theta$ using supervised learning to construct this cognitive map without external interaction. Figure 2 shows an illustrative example of the process. Our investigation focuses on whether autoregressive generation of the cognitive map and plan can induce extrapolability in path planning within Textualized Gridworld.

## 4 EXPERIMENTAL DESIGN

Building upon our discussion in Section 2.2, we conduct experiments using the Textualized Gridworld environment (Brown, 2015). Our experimental design encompasses two main planning scenarios and explores various cognitive map constructions and baselines. Especially, along three key dimensions: planning scenarios, baseline vs. cognitive map approaches, and map construction methods. Each dimension offers distinct variants, allowing for a comprehensive exploration of path planning capabilities in language models.

**Planning scenarios**    We investigate two corresponding types of planning analyses, optimal and reachable planning (See Figure 3), where

- **Optimal path planning:** The model generates the entire optimal plan in a single-turn manner.

- **Reachable path planning:** The model produces a reachable (not necessarily optimal) plan through multiple interactions with the given environment.

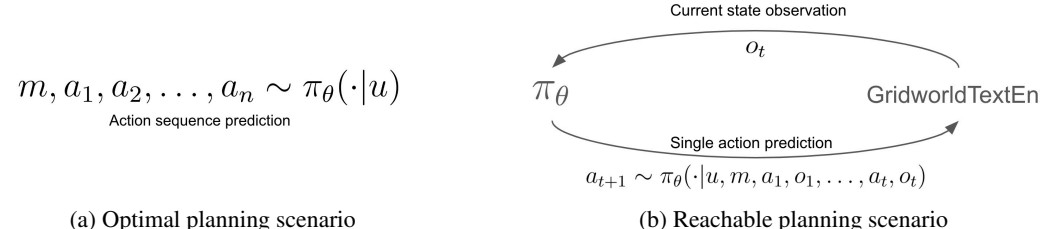

(a) Optimal planning scenario        (b) Reachable planning scenario

Figure 3: Optimal planning agent plans the whole action sequence including the CoT($m$) at once, occurring in a single turn (a). On the other hand, reachable planning agent iteratively plans and updates its strategy through multiple turns of interaction with the environment, occurring through multiple turns (b).

For multi-turn scenarios, we set a maximum of 200 steps per environment. Each observation provides the current state and possible non-deadend moves. For example, if current state is (11, 4) and there are pits or walls in up and down, the observation would be "Current:\n(11, 4)\nPossible:\n(10, 4)\nleft\n(12, 4)\nright". See Appendix C for a detailed explanation.

**Baseline vs. Cognitive map approaches**    We compare two baseline methods against two cognitive map variants, where baselines are

- NONE: Implicit learning without verbalization.

- COT: Explicitly learn the verbalized backward trace as a Chain of Thought (CoT) manner (Wei et al., 2023) (e.g., "(3, 2)up\n(3, 1)right\n...\n(0, 0)" for the case in Figure 2).

and cognitive map variants are

- MARK: Explicitly marks deadend states (e.g., "(0, 1) up (0, -1) *deadend* (1, 0) right (-1, 0) *deadend*" for propagation in the start state in Figure 2).

- UNMARK: Verbalizes all actions without marking deadends (e.g., "(0, 1) up (0, -1) down (1, 0) right (-1, 0) left" for propagation in the start state in Figure 2).

**Map construction methods**    LAMBADA (Kazemi et al., 2023) showed that "backward chaining" of language models can help its planning ability. Inspired by that, we explore two construction approaches:

- FWD: Forward construction, builds the map from start to goal.

- BWD: Backward construction, constructs the map from goal to start, with:
    - Sampling: Identical to FWD ($S$).
    - Propagation: Uses reverse transition $T^{-1}(s, a) \in \mathcal{S}$ for state $s$ and action $a$.
    - Backtracking: Traces path from start to goal using $T(s, a) \in \mathcal{S}$.

Note: The NONE baseline is identical for both FWD and BWD constructions.

**Experimental summary**    In total, our experimental design encompasses 2 planning scenarios (Optimal and Reachable), 4 approach variants (2 Baselines, 2 Cognitive Map) and 2 map construction methods (FWD and BWD). This results in 16 distinct experimental configurations ($2 \times 4 \times 2$), with the exception of NONE baseline, which is identical for both construction methods, reducing the total to 14 unique experiments. These comprehensive experiments allow us to evaluate the effectiveness of cognitive maps in enhancing language models' path planning capabilities, compare against baseline methods, and assess the impact of forward versus backward reasoning in complex spatial tasks. For detailed examples and complete constructions, please refer to Appendices D.1 and D.2.

## 5 EXPERIMENTAL RESULTS

**Extrapolation holds only for cognitive map**  To evaluate extrapolation capabilities, we trained our model on environments of up to $10 \times 10$ cells and tested on larger environments up to $20 \times 20$ cells. Our results in Table 1 demonstrate that cognitive maps significantly enhance path planning abilities on these larger, unseen environments. While baseline approaches (NONE and COT) failed to generalize beyond the training domain, the cognitive map architecture successfully extrapolated to larger spaces. For reachability planning specifically, both implicit and explicit baselines showed limited extrapolation capacity, particularly in environments where one dimension remained small. Notably, successful extrapolation cases typically exhibited complexity levels within the bounds of the training data's maximum complexity (Figure 5), suggesting that problem complexity, rather than raw environment size, may be the key determinant of extrapolation performance.

**Cognitive map vs. Baselines**  Table 2 compares optimal and reachable planning rates across experiments. For optimal planning, experiment with UNMARK performed best (76.5% for BWD, 61.8% for FWD). On the other hand, for reachable Planning: MARK excelled (88.5% for BWD, 85.4% for FWD).

Both finding implies that cognitive maps significantly improved performance for both optimal and reachable planning,

- Up to 57.5% boost in optimal planning and 56.4% in reachable planning vs. NONE
- Up to 50% improvement in optimal planning and 54.6% in reachable planning vs. COT

**Enhanced reachability with cognitive map**  We also observe that reachable planning shows substantial improvements over optimal planning in most configurations. For example, MARK w.o. BACKTRACK in FWD construction more than doubled its score (42.3% to 85.2%). This suggests that cognitive maps, while designed for optimal planning, also significantly enhance reachable planning capabilities. We can infer that even though the initial plan may be wrong, cognitive map can modify its initial plan in an online manner with the guidance of current observation.

**Benefit of thinking backward**  BWD cognitive map construction consistently outperformed FWD in both optimal and reachable planning. This aligns with LAMBADA's findings (Kazemi et al., 2023) on the benefits of backward chaining in reasoning tasks.

For detailed breakdowns across world sizes and experimental variants, see Appendices E.5 and E.6.

## 6 ADDITIONAL ANALYSIS AND DISCUSSION

Our experiments with cognitive maps in language models for path planning tasks reveal several intriguing insights that extend beyond mere performance improvements. In this section, we delve into the implications of our findings and discuss their broader relevance to the field of AI and cognitive science.

### 6.1 COGNITIVE MAP FOR LANGUAGE MODELS RESEMBLING HUMAN COGNITION

Aside from extrapolability, our design of cognitive maps exhibits two key characteristics that closely align with human cognition: **structured mental representation for planning** and **rapid adaptation during training**.

**Evidence from human planning behavior**  While the direct connection between explicit decision trees and the mental representation of cognitive maps remains elusive, eye-movement studies of human maze-solving have revealed crucial insights into human planning behavior (Kadner et al., 2023). These studies demonstrate that humans engage in mental simulation through gaze patterns that closely mirror the maze's structural elements. Additionally, humans display sophisticated search strategies, dynamically balancing between depth and breadth-first approaches based on the environment's complexity.

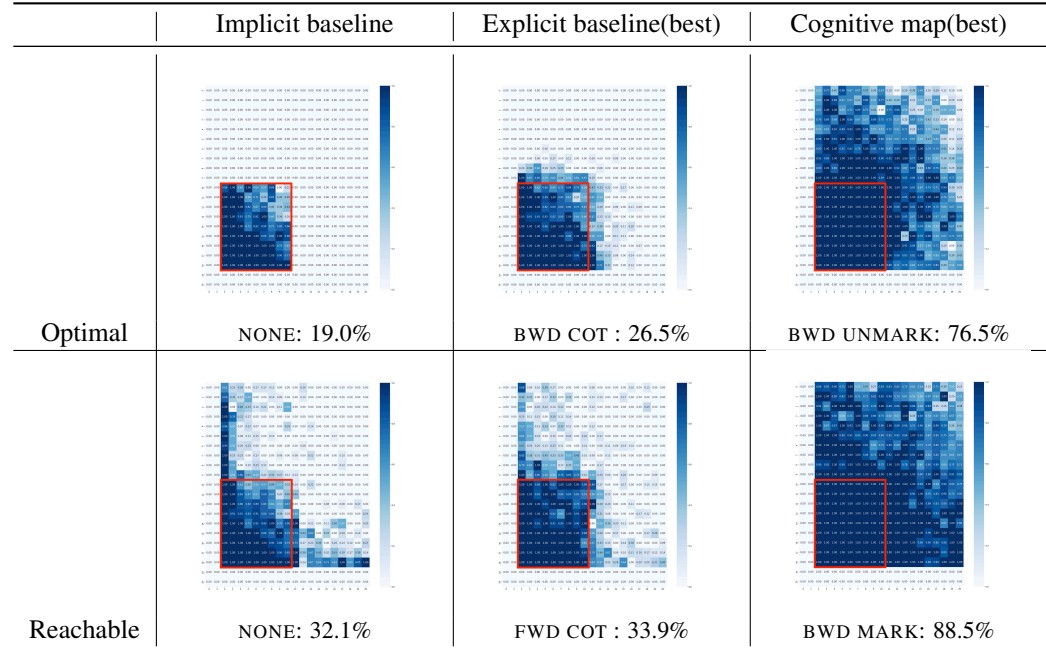

|  | Implicit baseline | Explicit baseline(best) | Cognitive map(best) |
|---|---|---|---|
| Optimal | NONE: 19.0% | BWD COT : 26.5% | BWD UNMARK: 76.5% |
| Reachable | NONE: 32.1% | FWD COT : 33.9% | BWD MARK: 88.5% |

Table 1: Qualitative comparison of reachable and optimal plan generation rate between baselines and our methods. The degree of the darkness at (x, y) coordinate of each plot tells the performance of the corresponding model in Gridworld of size x×y. The red box denotes the boundary of the training data. We provide visualization of all the experiments in Appendix E.

Our implementation of the cognitive map for language models is designed to exhibit similar patterns, constructing mental representations before taking actions and adapting its planning depth and breadth based on environmental complexity during the inference stage. While our implementation undoubtedly represents a simplified version of human cognitive maps, this parallel structure allows us to capture key aspects of human-like planning and problem-solving capabilities.

**Rapid adaptation with cognitive maps**   Beyond characteristics in the inference stage, we observe that employing cognitive maps leads to rapid convergence during training, another hallmark of human cognition (Lake et al., 2017). Models using cognitive maps demonstrate quick learning and adaptation to new scenarios. This fast learning capability suggests that the global representation provided by cognitive maps enables more efficient transfer of knowledge across similar but distinct problem spaces. We present detailed results in Appendix E.1.

## 6.2 DISTINGUISHING COGNITIVE MAPS FROM CONVENTIONAL COT: FEW-SHOT EXPERIMENTS

Our results in Section 5 demonstrate that models using cognitive maps could extrapolate to larger, unseen environments, while conventional approaches could not. This stark difference prompts a fundamental question: What distinguishes cognitive maps from conventional Chain of Thought (CoT) approaches? To investigate this, we conducted few-shot prompting experiments with various language models, including Llama-3-70B, GPT-4o, and GPT-o1, using prompts for NONE (baseline), COT, and COGNITIVE MAP approaches. (Refer Appendix E.2 for detailed descriptions of few-shot experiment) Our experiments revealed two crucial findings:

**Unpromptable nature of cognitive maps**   None of the tested foundation models could produce meaningful results for the cognitive map approach through few-shot prompting, regardless of their size or architecture, and even with increasing numbers of examples (0-shot to 4-shot). This observation stands in stark contrast to conventional Chain of Thought (CoT), which can typically be induced through prompting (Wei et al., 2023). The consistent failure to prompt for cognitive map reasoning

|  | Implicit baseline | Explicit baseline | Cognitive map | |
|---|---|---|---|---|
| Optimal | NONE | CoT | MARK | UNMARK |
| BWD | 0.190 | 0.265 | 0.705 | **0.765** |
| FWD | 0.190 | 0.252 | 0.585 | 0.618 |
| Reachable | NONE | CoT | MARK | UNMARK |
| BWD | 0.321 | 0.287 | **0.885** | 0.724 |
| FWD | 0.321 | 0.339 | 0.854 | 0.816 |

Table 2: Optimal and reachable rate of generated plans via single- and multi-turn settings: The first two columns(NONE and BACKTRACK) are the baselines for imitation-based learning, and the rest are different design choices of constructing the cognitive map. Also BWD constructs the map starting from the goal state, while FWD starts from the start state. See Appendix D for actual prompts.

suggests that this approach requires capabilities not inherently present in current large language models' training.

**Unique performance of GPT-o1 with baseline prompting**    While the exact implementation details, CoT mechanisms, and training dataset of GPT-o1 remain undisclosed, its unique performance with NONE prompting (Up to 38% with 4-shot) suggests intriguing possibilities. One explanation could be that GPT-o1 has been trained or fine-tuned to refine its reasoning process, explore alternative strategies, and iteratively recognize and correct its mistakes. This could make it more adept at reasoning that forms mental representations of spatial relationships, even without explicit prompting for cognitive maps.

However, it is unclear whether GPT-o1's reasoning capabilities align with the core principles of cognitive maps, particularly their extrapolability to unseen, complex environments. Without direct evidence of its performance in such settings, we can only speculate whether GPT-o1 represents a step toward generalized cognitive maps for language models. These findings hint at the potential for specialized training approaches to enable models to construct and utilize cognitive maps effectively, but further investigation is needed to draw definitive conclusions.

### 6.3 COMPARISON WITH EXPLORATION-BASED PLANNING

How does our cognitive map approach compare to exploration-based planning methods? Recent works have introduced various exploration strategies for language models, including Tree of Thoughts (ToT) (Yao et al., 2023), which combines language model sampling with value estimation for tree construction and traversal. Additional methods enhance exploration through techniques like Monte Carlo Tree Search (MCTS) (Kocsis & Szepesvári, 2006; Hao et al., 2023; Zhou et al., 2023). However, two significant challenges emerge in direct comparisons.

First, exploration-based methods are inherently constrained to reachable planning scenarios, where decisions occur incrementally during inference. This fundamental limitation prevents direct comparisons with optimal planning approaches like our cognitive maps. Second, our analysis of conventional LLMs' sampling behavior reveals consistent overconfidence in specific directions, leading to ineffective tree structure generation and exploration in Gridworld environments. This behavioral limitation undermines the effectiveness of methods like ToT (Yao et al., 2023) and RAP (Hao et al., 2023) in tasks requiring systematic search and robust spatial reasoning.

To enable meaningful comparison, we developed an exploration-based baseline where language models were specifically trained to perform Depth-First Search (DFS) for reachability analysis. This controlled approach allowed us to evaluate structured exploration under conditions comparable to our cognitive map method, while focusing on reachability analysis in planning tasks. Our experiments show that DFS-based exploration achieves higher reachability rates (94.5%) compared to our cognitive map approach (88.5%) in reachable planning scenarios (Table 11). However, cognitive maps demonstrate superior efficiency in path optimality. While DFS requires $O(n^2)$ steps for paths of

optimal length $n$, our method achieves near-optimal performance ($O(n)$) even in reachable planning settings (Figure 10) (details in Appendix E.3).

These findings highlight a crucial trade-off: while exploration-based methods achieve higher success rates through exhaustive tree search in reachable planning scenarios, cognitive maps enable more efficient planning through structured environmental representations as a CoT. This efficiency difference illuminates two fundamental planning paradigms:

- **Offline Planning via Cognitive Maps:** Cognitive maps construct mental representations during the CoT stage, enabling complete decision process simulation before action.
- **Online Planning through Exploration:** Exploration-based planning utilizes active exploration for incremental spatial understanding and adaptive plan refinement.

Both approaches play vital roles in human cognition, serving complementary functions. Offline planning facilitates high-level strategy formation and global understanding, while online planning enables real-time adjustments and responsive problem-solving in dynamic environments.

## 7 CONCLUSION

Our work serves as a proof-of-concept for training language models with enhanced spatial reasoning capabilities through targeted supervision of cognitive map construction. While we do not claim to replicate exact human learning processes, our results demonstrate that enabling structured mental representations through cognitive maps leads to superior extrapolation abilities in path planning tasks.

Our findings raise two important considerations for future research directions:

- **Development of general-purpose cognitive maps:** Our experiments reveal the unpromptable nature of cognitive map reasoning, indicating that current pre-training approaches are insufficient for developing such cognitive capabilities. While our implementation demonstrates success in the Gridworld domain, extending these principles to more general tasks and abstract problem spaces requires further investigation. Human cognitive maps are significantly more complex than our current implementations, encompassing not only spatial reasoning but also abstract concept spaces and relational knowledge. The unique performance of models like GPT-o1 suggests that alternative training paradigms, such as reinforcement learning combined with large-scale Chain of Thought (CoT) data augmentation, might offer a more organic path to acquiring cognitive map-like representations. Additionally, exploring techniques for generating diverse reasoning traces through tree search could provide valuable insights into embedding these capabilities more effectively into foundation models.
- **Integration of different planning paradigms:** Our work highlights the complementary nature of different planning paradigms. While this study primarily examines the offline planning capabilities of cognitive maps, our comparison with exploration-based methods reveals interesting trade-offs between planning efficiency and success rates. Future research could explore hybrid approaches that integrate the strengths of both cognitive maps and online exploration. Such frameworks could bridge the gap between offline and online reasoning while more closely approximating the dual planning modes observed in human cognition.

Despite these open challenges, our work makes several significant contributions to the field. We systematically demonstrate that a specific form of CoT prompting, structured as a cognitive map, can enable extrapolation in spatial reasoning tasks. Our implementation exhibits characteristics that parallel human cognitive processes, including structured mental representations and rapid adaptation during training. The unpromptable nature of cognitive map reasoning suggests fundamental limitations in current pre-training approaches and highlights the need for specialized training schemes.

Looking ahead, this work opens new avenues for developing more sophisticated cognitive architectures in language models. While our current implementation represents a simplified version of human cognitive maps, it provides a foundation for understanding how structured mental representations can enhance planning and reasoning capabilities. Future work in this direction could lead to more robust and adaptable AI systems capable of human-like reasoning across diverse problem domains.

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

## A RELATED WORKS

### A.1 PLANNING IN COGNITIVE SCIENCE

Dual-process theories of cognition distinguish between System 1 and System 2 reasoning, where System 1 involves fast, automatic, and intuitive thinking, and System 2 involves slow, deliberate, and analytical thought (Kahneman, 2011). Especially, System 2 cognition for human typically engages in iterative mental simulations, refining steps until reaching the goal while avoiding dead-end states. This process involves the formation and use of "cognitive maps," which are mental representations of spatial environments allowing for flexible navigation and planning. While traditionally associated with the hippocampus (O'Keefe & Nadel, 1978), cognitive maps involve a network of brain regions. The prefrontal cortex plays a crucial role in utilizing these maps for planning and decision-making (Daw et al., 2005), the hippocampus is central to their formation, and regions like the parietal cortex contribute to spatial processing. This distributed neural system allows for deliberate, complex decision-making (Doll et al., 2012), enabling humans to adapt learned behaviors to novel, complex environments.

One of the key characteristics of human cognition is extrapolation. There are multiple evidences across cognitive science domains proving that humans are able to solve problems in larger, more complex environments that were not encountered during the initial learning phase (Yousefzadeh & Mollick, 2021; Fellini & Morellini, 2011; Stojic et al., 2018). This capability allows humans to apply learned knowledge to novel situations, demonstrating a remarkable level of generalization.

### A.2 PLANNING IN LANGUAGE MODEL DOMAIN

Language models are primarily trained using next-token prediction, which facilitates model-free planning by allowing them to generate text based on learned patterns and associations from vast amounts of data. This training approach enables language models to perform zero-shot planning in various domains, such as general planning and code generation, without requiring task-specific training Chen et al. (2021). Language models excel at pattern matching and leverage their learned world model to plan and generate coherent text in given tasks Brown et al. (2020).

Despite their successes, language models often struggle with planning tasks that require extensive foresight and detailed reasoning. Studies shows that language models perform poorly on tasks requiring robust planning and complex decision-making, highlighting the limitations of model-free approaches (Xie et al., 2024; Valmeekam et al., 2022).

One way to inject planning capability for language models is by imitating the "thought" of the reasoning as a part of the generation. We call it as Chain of Thought (CoT)(Wei et al., 2023) methodology, and it has demonstrated significant improvements in language models' reasoning and planning abilities. They involve guiding the model through a series of chains generated by human or ground truth that simulates step-by-step reasoning, enhancing the model's decision-making processes via few-shot demonstration or fine-tuning. They imitate ground truth reasoning demonstrations to improve the performance in tasks requiring complex reasoning (Nye et al., 2021; Yao et al., 2022).

One limitation of conventional CoT is that there is lack of opportunity to explore a wide range of world states. To address this, exploration-based methods have been designed to enable language models to explore more effectively. One such example is Tree of Thought (ToT) (Yao et al., 2023), which leverages language models' knowledge to sample child nodes and simultaneously evaluate the value of the current state to structure and search the tree structure at the same time. Also one can enhance the searching by using Monte Carlo Tree Search (MCTS) algorithm (Kocsis & Szepesvári, 2006) to explore states and update their value (Hao et al., 2023; Zhou et al., 2023). These exploration-based planning methods enhance the model's ability to reach the goal, with a perfect probability if we have infinite time and inference cost. Methods such as Searchformer (Lehnert et al., 2024) experimentally shows that imitating A* algorithm rather than human demonstration is also helpful when planning. The searching algorithm conducts tree search via heuristic cost function, yet they show that the policy of the model can further imply optimal solutions. It implies that the language models can somewhat "extract" optimal plans from heuristic searching algorithms.

## A.3 EXTRAPOLATION INTO MORE COMPLEX ENVIRONMENTS

Extrapolation to complex environments remains a significant challenge for language models in planning tasks. Despite their expressive power, conventional approaches struggle with generalization to more intricate scenarios:

- Imitation-based planning, even with advanced models like GPT-4 (OpenAI et al., 2024), fails to generalize effectively to compositional tasks such as multiplication, Einstein's puzzle (Dziri et al., 2024), or Blocksworld puzzle (Valmeekam et al., 2022; Stechly et al., 2024) when faced with extrapolated data.

- Chain of Thought (CoT) methods, while enhancing reasoning capabilities (Merrill & Sabharwal, 2024; Feng et al., 2024), show limited effectiveness in extrapolation. Theoretical work suggests that extensive CoT demonstrations are needed for System 2 reasoning problems, yet this often proves insufficient for true extrapolation (Peng et al., 2024).

- Tree search methods in language models also enhances the planning ability of language models (Yao et al., 2023; Hao et al., 2023; Zhou et al., 2023) but their extrapolation potential remains understudied. Existing research primarily focuses on general improvements in planning or reasoning without specifically addressing extrapolation to more complex environments (Daw et al., 2005).

These limitations contrast sharply with the adaptability observed in human cognitive planning, highlighting a critical gap in current language model applications. The challenge lies in developing approaches that can effectively bridge this gap, enabling more robust planning ability.

## A.4 EVALUATING COGNITIVE MAPS IN LANGUAGE MODELS

Recent researches focus on evaluating the spatial understanding and cognitive mapping capabilities of LLMs. This section discusses key findings from studies that explore how well LLMs can represent and reason about spatial relationships, which are crucial for cognitive mapping.

- Spatial Understanding: Recent study (Yamada et al., 2024) shows that LLMs can implicitly capture aspects of spatial structures despite being trained only on text. These models were tested on tasks involving navigation through various grid structures such as squares, hexagons, and trees. The results indicated variability in performance, suggesting that while LLMs exhibit some spatial reasoning capabilities, there is significant room for improvement.

- Cognitive Mapping and Planning: Recent study (Momennejad et al., 2023) highlights the challenges LLMs face in forming cognitive maps necessary for effective planning. Although these models can handle basic spatial tasks, their ability to generalize and plan in more complex environments remains limited. This limitation is evident when comparing their performance to human benchmarks, where non-expert humans often outperform LLMs in spatial reasoning tasks.

- Benchmarking and Error Analysis: Comprehensive benchmarking studies (Li et al., 2024; Xu et al., 2024) systematically evaluate LLMs on spatial tasks. These studies use multi-task evaluation datasets to assess the models' performance across different spatial reasoning challenges. Error analyses reveal that LLMs' mistakes often stem from both spatial and non-spatial factors, indicating areas where further refinement is needed.

These findings highlight the importance of developing more advanced training techniques and benchmarks to improve the cognitive mapping abilities of LLMs, enabling them to better emulate human-like planning and reasoning. Building on these studies, we evaluate the challenges language models face in demonstrating spatial reasoning (Gridworld), particularly in extrapolated environments that are unseen during training. However, our approach differs in that we propose designing cognitive maps specifically for path planning as a potential solution to address these limitations.

# B EXPERIMENTAL DETAILS

## B.1 INPUT DETAIL

Table 3 describes a sample input of the model, describing the instruction of the Gridworld and the specific world information.

| Common Prompt | human: You are given a rectangular gridworld, where you can move up, down, left, or right as long as each of your x, y coordinates are within 0 to the x, y size of the grid. If you move up, your y coordinate increases by 1. If you move down, your y coordinate decreases by 1. If you move left, your x coordinate decreases by 1. If you move right, your x coordinate increases by 1.\n\nYou will interact with the gridworld environment to reach the goal state, while avoiding the pit and the wall. You cannot move through the wall or move outside the grid. If you fall into the pit, you lose. If you reach the goal, you win. For each of your turn, you will be given the possible moves.\n\nYou should respond your move with either one of 'up', 'down', 'left', or 'right'. gpt: OK human: Grid is from (0, 0) to (3, 2). Goal: (3, 2)\nCurrent: (0, 0)\nThe pit is at (3, 0). The wall is at (1, 1), and (2, 2).\nCurrent:\n(0, 0)\nPossible:\n(0, 1)\nup\n(1, 0)\nright |
|---|---|

Table 3: The prompt for all cases

## B.2 EXPERIMENTAL SETUP DETAILS

We utilize Llama-3-8B model[2] with a maximum token length of 8,192 per instance. We train the model for 1 epoch with 50K samples of size $10 \times 10$ at largest. After training, we test the model with 3K samples with each grid being $20 \times 20$ at largest.

We use one 8 Nvidia A100 node for both training and inference. For the training steps, we use FSDP framework (Zhao et al., 2023) and cosine annealing learning rate scheduler (Loshchilov & Hutter, 2017) for 1 epoch. We utilize bfloat16 floating-point format and a warmup ratio of 0.03. We set the weight decay as 0. We set the batch size of 2 for each GPUs, so the effective batch size is 16 per step. We train each model for $50000/16 = 3125$ steps. For inference, we use VLLM framework (Kwon et al., 2023).

While exploring the pure planning ability of the language model, we did not want the model to refuse to explore extrapolated data only because it has never seen the coordinate. To handle the bias, we adjust the starting position of the grid using uniform sampling while ensuring that the entire grid, with its x and y coordinates, fits within the range of 0 to 19. This method, illustrated in Figure 4, minimizes bias related to the unseen x and y coordinates by randomizing the starting point within the defined bounds.

---

[2]https://llama.meta.com/llama3/

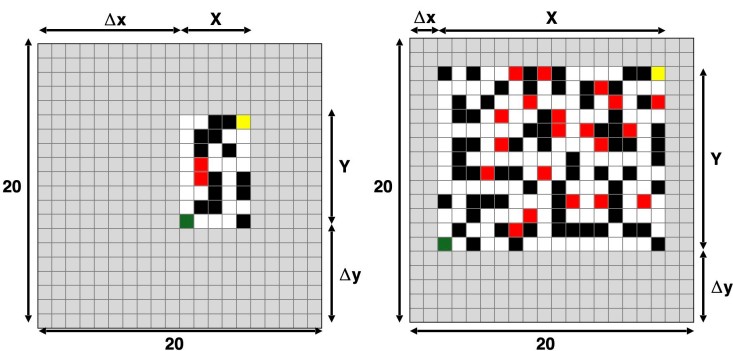

Figure 4: Visualization of configuring Gridworld instance for train(left) and test(right) dataset. To evaluate the extrapolation ability, we set the size of the grid as $X, Y \sim Unif(2, 10)$ for train and $X, Y \sim Unif(2, 20)$ for test. Also we set the starting point of the grid as $\Delta x \sim Unif(0, 20 - X), \Delta y \sim Unif(0, 20 - Y)$ for both train and test.

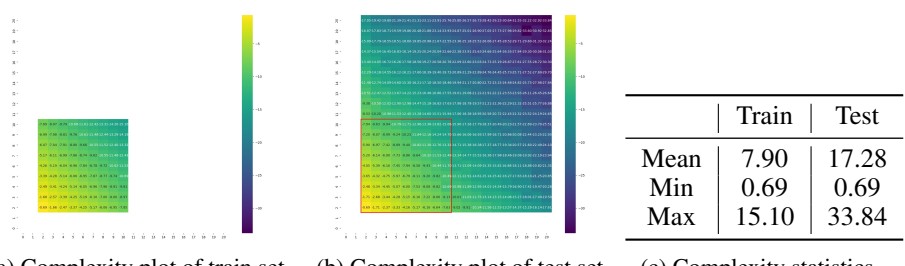

| | Train | Test |
|---|---|---|
| Mean | 7.90 | 17.28 |
| Min | 0.69 | 0.69 |
| Max | 15.10 | 33.84 |

(a) Complexity plot of train set    (b) Complexity plot of test set    (c) Complexity statistics

Figure 5: Complexity analysis of Gridworld environments. Each heatmap (a, b) shows the average complexity (Equation 1) of Gridworlds of size $x \times y$, where the darkness at each $(x, y)$ coordinate represents the mean complexity. **(a)** depicts the training set, and **(b)** shows the test set, with the red box (from $(2, 2)$ to $(10, 10)$) indicating the training boundary. **(c)** provides summary statistics of complexity for both train and test sets. Details on train/test dataset statistics can be found in Appendix B.4.

## B.3 COMPLEXITY ANALYSIS

The complexity of a grid environment is defined as the negative log probability of a random policy successfully following the optimal path. Here, the random policy does not blindly select actions but instead chooses uniformly from the set of valid actions at each state—excluding those that would lead to invalid moves, such as bumping into walls or falling into pits. For an environment with $L$ as the length of the optimal path and $A_s$ as the number of valid actions at state $s$, the complexity $C$ can be computed as:

$$C = -\log\left(\prod_{s=1}^{L} \frac{1}{A_s}\right) = \sum_{s=1}^{L} \log(A_s) \tag{1}$$

Here, $L$ corresponds to the number of steps in the optimal path, and $A_s$ varies depending on the state $s$, reflecting the number of valid actions available at that point. As grid dimensions increase, $L$ grows due to more complex environments with additional obstacles and decision points, leading to higher complexity values. Figure 5 presents complexity statistics for both train and test dataset.

## B.4 INPUT/OUTPUT STATISTICS

Figures 6 and 7 present detailed statistics for input and plan token lengths, respectively. Similar to the complexity analysis discussed in Section 2.2, notable discrepancies exist between the train and

test datasets, emphasizing the extrapolation challenges posed by larger and more complex Gridworld environments.

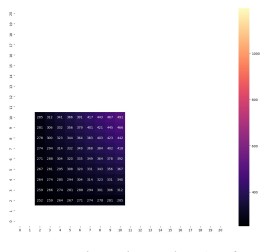
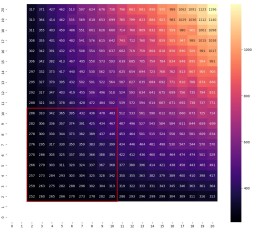

| Metric | Train | Test |
|--------|-------|------|
| Mean | 331.98 | 531.09 |
| Min | 252 | 252 |
| Max | 631 | 1323 |

(a) Input token lengths (train set)    (b) Input token lengths (test set)    (c) Input token statistics

Figure 6: Input length analysis of Gridworld environments. Heatmaps (a, b) represent the average input token length for Gridworlds of size $x \times y$, where the darkness at each $(x, y)$ coordinate indicates the mean token length. **(a)** shows the training set, and **(b)** shows the test set. The red box (from $(2, 2)$ to $(10, 10)$) outlines the training boundary. Summary statistics are provided in (c).

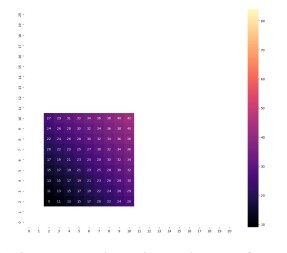
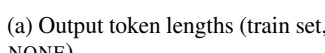
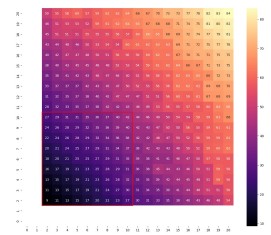
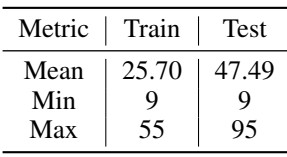

| Metric | Train | Test |
|--------|-------|------|
| Mean | 25.70 | 47.49 |
| Min | 9 | 9 |
| Max | 55 | 95 |

(a) Output token lengths (train set, NONE)    (b) Output token lengths (test set, NONE)    (c) Output token statistics, NONE

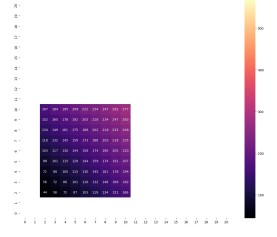
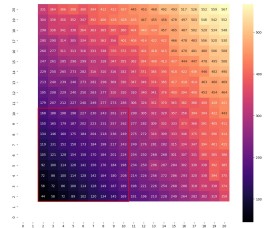
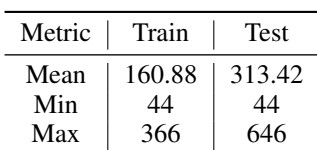

| Metric | Train | Test |
|--------|-------|------|
| Mean | 160.88 | 313.42 |
| Min | 44 | 44 |
| Max | 366 | 646 |

(d) Output token lengths (train set, BWD COT)    (e) Output token lengths (test set, BWD COT)    (f) Output token statistics (BWD COT)

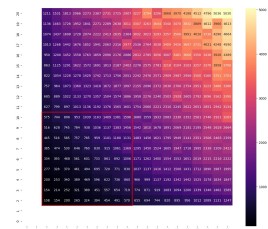

| Metric | Train | Test |
|--------|-------|------|
| Mean | 666.31 | 1146.47 |
| Min | 108 | 108 |
| Max | 2130 | 6134 |

(g) Output token lengths (train set, BWD UNMARK)    (h) Output token lengths (test set, BWD UNMARK)    (i) Output token statistics (BWD UNMARK)

Figure 7: Output length analysis of Gridworld environments. Heatmaps (a, b) represent the average plan token length for Gridworlds of size $x \times y$. The darkness at each $(x, y)$ coordinate indicates the mean token length. **(a, b)** depict the direct path approach, **(d, e)** show Chain of Thought (CoT) reasoning, and **(g, h)** illustrate cognitive map reasoning. The red box (from $(2, 2)$ to $(10, 10)$) outlines the training boundary. Summary statistics for each approach are provided in (c, f, i).

## C  PLANNING TASK INTERACTING WITH WORLD MODEL

Planning task with environment feedback in language model can be formalized as a Markov decision process with instruction space $\mathcal{U}$, state space $\mathcal{S}$, action space $\mathcal{A}$, observation space $\mathcal{O}$, metric space $\mathcal{C}$, transition function $T : \mathcal{S} \times \mathcal{A} \to \mathcal{S}$, and metric function $R : \mathcal{U} \times (\mathcal{S} \times \mathcal{A})^n \to \mathcal{C}$ where $n$ is the trajectory length. Note that for language model domain, $\mathcal{U}$, $\mathcal{A}$, and $\mathcal{O}$ are given as sequences of language tokens.

Given an instruction $u \in \mathcal{U}$, the model $\theta$ first generates the action $a_1 \sim \pi_\theta(\cdot|u) \in \mathcal{A}$ according to its policy $\pi_\theta$. For each state $s_t \in S$ and its observation $o_t \in \mathcal{O}$, the agent generates the corresponding action in the $t+1$ step $a_{t+1} \sim \pi_\theta(\cdot|u, a_1, o_1, \ldots, a_t, o_t) \in \mathcal{A}$, which concludes to a new state $s_{t+1} = T(s_t, a_t)$ and its observation $o_{t+1}$. The interaction loop repeats until the task has terminated for some reason(succeeded, failed, number of steps exceeded maximum value, etc.), and the action trajectory is denoted as:

$$e = (u, a_1, o_1, \ldots, o_{n-1}, a_n, o_n) \sim \pi_\theta(e|u),$$

$$\pi_\theta(e|u) = \prod_{j=1}^n \pi_\theta(a_j|u, a_1, o_1, \ldots, o_j),$$

Finally, we define the (action, state) trajectory $k = ((a_1, s_1), \ldots, (a_n, s_n))$ accordingly. The final reward is computed based on the metric function $R(u, k)$.

Note that we can apply reasoning before deciding the action with so-called a "thinking" process. Namely, we have a thinking space $\mathcal{M}$(of language subset) which generates $m \sim \pi_\theta(\cdot|u) \in \mathcal{M}$, then generate $a_1 \sim \pi_\theta(\cdot|u, m) \in \mathcal{M}$ upon the generated thought.

There are two types of planning tasks:

**Optimal planning analysis: single-turn setting**   Optimal planning evaluates the model's ability to generate an optimal plan without further observations. The model generates the entire action sequence in a single turn: $a_1, a_2, \ldots, a_n \sim \pi_\theta(\cdot|u, m) \in \mathcal{A}^n$. We define success as generating the optimal path $k^*$, which is the shortest successful trajectory $\arg\min_{k \in \{k|r(u,k)=\text{success}\}} |k|$. Since we ensure there is only one possible $k^*$, we only need to check whether the generated trajectory is $k^*$. Now we can define $R(u, k)$ as follows:

- $R(u, k) = \textsc{success}$ if $k = k^*$
- $R(u, k) = \textsc{fail}$ otherwise

**Reachable planning analysis: multi-turn setting**   Reachable planning investigates whether the model can produce a reachable (not necessarily optimal) plan in a multi-turn setting. The model generates actions sequentially, considering previous actions and observations: $a_{t+1} \sim \pi_\theta(\cdot|u, m, a_1, o_1, \ldots, a_t, o_t) \in \mathcal{A}$. We define success as reaching the goal state, with failure cases including reaching a deadend, exceeding the maximum number of steps, or generating invalid actions. Especially, given the instruction $u$ and generated (action, state) trajectory $k$, we define $R(u, k)$ as follows:

- $R(u, k) = \textsc{success}$ if $k[-1][1] = goal$
- $R(u, k) = \textsc{deadend}$ if $k[-1][1] \in P$
- $R(u, k) = \textsc{max step}$ if $|k| > max$
- $R(u, k) = \textsc{invalid}$ if $\exists a \in k[:][0] \mid a \notin A$

## D  COGNITIVE MAP DESCRIPTION

### D.1  COGNITIVE MAP EXAMPLE: FWD

Table 4 describes a sample of FWD cognitive map construction for each experiment.

| Design choice | Cognitive map example |
|---|---|
| NONE | |
| COT | Thought:\nStep 1:\nStep 2:\nStep 3:\nStep 4:\nStep 5:\nBacktrack:\n(3, 2)up\n(3, 1)\nright\n(2, 1)\nup\n(2, 0)\nright\n(1, 0)\nright\n(0, 0) |
| MARK W.O. ALL BACKTRACK | Thought:\nStep 1:\n(0, 1)\nup\n(1, 0)\nright\nStep 2:\n(0, 2)\nup\n(2, 0)\nright\nStep 3:\n(1, 2)\nright\n(2, 1)\nup\nStep 4:\n(3, 1)\nright\nStep 5:\n(3, 2)\nup |
| MARK W.O. ALL | Thought:\nStep 1:\n(0, 1)\nup\n(1, 0)\nright\nStep 2:\n(0, 2)\nup\n(2, 0)\nright\nStep 3:\n(1, 2)\nright\n(2, 1)\nup\nStep 4:\n(3, 1)\nright\nStep 5:\n(3, 2)\nup\nBacktrack:\n(3, 2)up\n(3, 1)\nright\n(2, 1)\nup\n(2, 0)\nright\n(1, 0)\nright\n(0, 0) |
| UNMARK W.O. BACKTRACK | Thought:\nStep 1:\n(0, 1)\nup\n(0, -1)\ndown\n(-1, 0)\nleft\n(1, 0)\nright\nStep 2:\n(0, 2)\nup\n(0, 0)\ndown\n(-1, 1)\nleft\n(1, 1)\nright\n(1, 1)\nup\n(1, -1)\ndown\n(0, 0)\nleft\n(2, 0)\nright\nStep 3:\n(0, 3)\nup\n(0, 1)\ndown\n(-1, 2)\nleft\n(1, 2)\nright\n(2, 1)\nup\n(2, -1)\ndown\n(1, 0)\nleft\n(3, 0)\nright\nStep 4:\n(1, 3)\nup\n(1, 1)\ndown\n(0, 2)\nleft\n(2, 2)\nright\n(2, 2)\nup\n(2, 0)\ndown\n(1, 1)\nleft\n(3, 1)\nright\nStep 5:\n(3, 2)\nup\n(3, 0)\ndown\n(2, 1)\nleft\n(4, 1)\nright |
| UNMARK | Thought:\nStep 1:\n(0, 1)\nup\n(0, -1)\ndown\n(-1, 0)\nleft\n(1, 0)\nright\nStep 2:\n(0, 2)\nup\n(0, 0)\ndown\n(-1, 1)\nleft\n(1, 1)\nright\n(1, 1)\nup\n(1, -1)\ndown\n(0, 0)\nleft\n(2, 0)\nright\nStep 3:\n(0, 3)\nup\n(0, 1)\ndown\n(-1, 2)\nleft\n(1, 2)\nright\n(2, 1)\nup\n(2, -1)\ndown\n(1, 0)\nleft\n(3, 0)\nright\nStep 4:\n(1, 3)\nup\n(1, 1)\ndown\n(0, 2)\nleft\n(2, 2)\nright\n(2, 2)\nup\n(2, 0)\ndown\n(1, 1)\nleft\n(3, 1)\nright\nStep 5:\n(3, 2)\nup\n(3, 0)\ndown\n(2, 1)\nleft\n(4, 1)\nright\nBacktrack:\n(3, 2)up\n(3, 1)\nright\n(2, 1)\nup\n(2, 0)\nright\n(1, 0)\nright\n(0, 0) |
| MARK W.O. BACKTRACK | Thought:\nStep 1:\n(0, 1)\nup\n(0, -1)\ncut\n(-1, 0)\ncut\n(1, 0)\nright\nStep 2:\n(0, 2)\nup\n(0, 0)\ncut\n(-1, 1)\ncut\n(1, 1)\ncut\n(1, 1)\ncut\n(1, -1)\ncut\n(0, 0)\ncut\n(2, 0)\nright\nStep 3:\n(0, 3)\ncut\n(0, 1)\ncut\n(-1, 2)\ncut\n(1, 2)\nright\n(2, 1)\nup\n(2, -1)\ncut\n(1, 0)\ncut\n(3, 0)\ncut\nStep 4:\n(1, 3)\ncut\n(1, 1)\ncut\n(0, 2)\ncut\n(2, 2)\ncut\n(2, 2)\ncut\n(2, 0)\ncut\n(1, 1)\ncut\n(3, 1)\nright\nStep 5:\n(3, 2)\nup\n(3, 0)\ncut\n(2, 1)\ncut\n(4, 1)\ncut |
| MARK | Thought:\nStep 1:\n(0, 1)\nup\n(0, -1)\ncut\n(-1, 0)\ncut\n(1, 0)\nright\nStep 2:\n(0, 2)\nup\n(0, 0)\ncut\n(-1, 1)\ncut\n(1, 1)\ncut\n(1, 1)\ncut\n(1, -1)\ncut\n(0, 0)\ncut\n(2, 0)\nright\nStep 3:\n(0, 3)\ncut\n(0, 1)\ncut\n(-1, 2)\ncut\n(1, 2)\nright\n(2, 1)\nup\n(2, -1)\ncut\n(1, 0)\ncut\n(3, 0)\ncut\nStep 4:\n(1, 3)\ncut\n(1, 1)\ncut\n(0, 2)\ncut\n(2, 2)\ncut\n(2, 2)\ncut\n(2, 0)\ncut\n(1, 1)\ncut\n(3, 1)\nright\nStep 5:\n(3, 2)\nup\n(3, 0)\ncut\n(2, 1)\ncut\n(4, 1)\ncut\nBacktrack:\n(3, 2)up\n(3, 1)\nright\n(2, 1)\nup\n(2, 0)\nright\n(1, 0)\nright\n(0, 0) |

Table 4: FWD cognitive map example for each experiment

D.2 COGNITIVE MAP EXAMPLE: BWD

Table 5 describes a sample of BWD cognitive map construction for each experiment.

| Design choice | Cognitive map example |
|---|---|
| NONE | |
| COT | Thought:\nStep 1:\nStep 2:\nStep 3:\nStep 4:\nStep 5:\nBacktrack:\n(0, 0)\nright\n(1, 0)\nright\n(2, 0)\nup\n(2, 1)\nright\n(3, 1)\nup\n(3, 2) |
| MARK W.O. ALL BACKTRACK | Thought:\nStep 1:\n(3, 1)\nup\nStep 2:\n(2, 1)\nright\nStep 3:\n(2, 0)\nup\nStep 4:\n(1, 0)\nright\nStep 5:\n(0, 0)\nright |
| MARK W.O. ALL | Thought:\nStep 1:\n(3, 1)\nup\nStep 2:\n(2, 1)\nright\nStep 3:\n(2, 0)\nup\nStep 4:\n(1, 0)\nright\nStep 5:\n(0, 0)\nright\nBacktrack:\n(0, 0)\nright\n(1, 0)\nright\n(2, 0)\nup\n(2, 1)\nright\n(3, 1)\nup\n(3, 2) |
| UNMARK W.O. BACKTRACK | Thought:\nStep 1:\n(3, 3)\ndown\n(3, 1)\nup\n(2, 2)\nright\n(4, 2)\nleft\nStep 2:\n(3, 2)\ndown\n(3, 0)\nup\n(2, 1)\nright\n(4, 1)\nleft\nStep 3:\n(2, 2)\ndown\n(2, 0)\nup\n(1, 1)\nright\n(3, 1)\nleft\nStep 4:\n(2, 1)\ndown\n(2, -1)\nup\n(1, 0)\nright\n(3, 0)\nleft\nStep 5:\n(1, 1)\ndown\n(1, -1)\nup\n(0, 0)\nright\n(2, 0)\nleft |
| UNMARK | Thought:\nStep 1:\n(3, 3)\ndown\n(3, 1)\nup\n(2, 2)\nright\n(4, 2)\nleft\nStep 2:\n(3, 2)\ndown\n(3, 0)\nup\n(2, 1)\nright\n(4, 1)\nleft\nStep 3:\n(2, 2)\ndown\n(2, 0)\nup\n(1, 1)\nright\n(3, 1)\nleft\nStep 4:\n(2, 1)\ndown\n(2, -1)\nup\n(1, 0)\nright\n(3, 0)\nleft\nStep 5:\n(1, 1)\ndown\n(1, -1)\nup\n(0, 0)\nright\n(2, 0)\nleft\nBacktrack:\n(0, 0)\nright\n(1, 0)\nright\n(2, 0)\nup\n(2, 1)\nright\n(3, 1)\nup\n(3, 2) |
| W.O. BACKTRACK | Thought:\nStep 1:\n(3, 3)\ncut\n(3, 1)\nup\n(2, 2)\ncut\n(4, 2)\ncut\nStep 2:\n(3, 2)\ncut\n(3, 0)\ncut\n(2, 1)\nright\n(4, 1)\ncut\nStep 3:\n(2, 2)\ncut\n(2, 0)\nup\n(1, 1)\ncut\n(3, 1)\ncut\nStep 4:\n(2, 1)\ncut\n(2, -1)\ncut\n(1, 0)\nright\n(3, 0)\ncut\nStep 5:\n(1, 1)\ncut\n(1, -1)\ncut\n(0, 0)\nright\n(2, 0)\ncut |
| MARK | Thought:\nStep 1:\n(3, 3)\ncut\n(3, 1)\nup\n(2, 2)\ncut\n(4, 2)\ncut\nStep 2:\n(3, 2)\ncut\n(3, 0)\ncut\n(2, 1)\nright\n(4, 1)\ncut\nStep 3:\n(2, 2)\ncut\n(2, 0)\nup\n(1, 1)\ncut\n(3, 1)\ncut\nStep 4:\n(2, 1)\ncut\n(2, -1)\ncut\n(1, 0)\nright\n(3, 0)\ncut\nStep 5:\n(1, 1)\ncut\n(1, -1)\ncut\n(0, 0)\nright\n(2, 0)\ncut\nBacktrack:\n(0, 0)\nright\n(1, 0)\nright\n(2, 0)\nup\n(2, 1)\nright\n(3, 1)\nup\n(3, 2) |

Table 5: BWD cognitive map example for each experiment

|  | w.o. ALL | | with ALL | |
| --- | --- | --- | --- | --- |
| Optimal | w.o. ALL BACKTRACK | w.o. ALL | UNMARK | MARK |
| BWD | 0.296 | 0.277 | **0.765** | 0.705 |
| FWD | 0.295 | 0.290 | **0.618** | 0.585 |
| Reachable | w.o. ALL BACKTRACK | w.o. ALL | UNMARK | MARK |
| BWD | 0.394 | 0.283 | 0.724 | **0.885** |
| FWD | 0.416 | 0.345 | 0.816 | **0.854** |

Table 6: Planning performance with ALL exclusion

|  | w.o. BACKTRACK | | with BACKTRACK | |
| --- | --- | --- | --- | --- |
| Optimal | w.o. MARK BACKTRACK | w.o. BACKTRACK | UNMARK | MARK |
| BWD | 0.406 | 0.423 | **0.765** | 0.705 |
| FWD | 0.528 | 0.516 | **0.618** | 0.585 |
| Reachable | w.o. MARK BACKTRACK | w.o. BACKTRACK | UNMARK | MARK |
| BWD | 0.739 | 0.852 | 0.724 | **0.885** |
| FWD | 0.672 | 0.624 | 0.816 | **0.854** |

Table 7: Planning performance with BACKTRACK exclusion

# E  ADDITIONAL RESULTS

## E.1  RAPID ADAPTATION

We experiment speed of the language model learning robust BWD MARK cognitive map construction by both watching its performance in reachability setting and its loss curve. As shown in Figure 8a, the success rate of the model converges at the early stage(79.13% at step 500, 74.5% at step 625), which implies that it is easy for a model to learn how to generate a robust cognitive map. We also observe that the loss curve of the model converges to 0 much faster than other baseline methods(Figure 8b). **It suggests that the language model rapidly learns how to construct the cognitive map.**

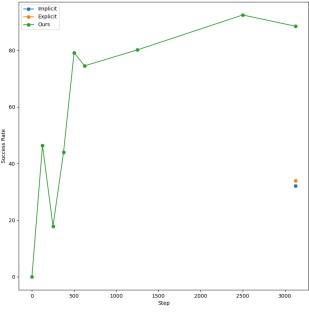

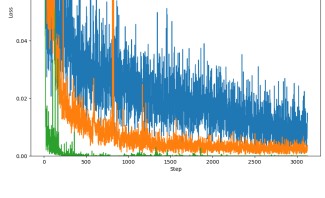

(a) Success rate comparison  (b) Loss curve comparsion

Figure 8: **(a)** and **(b)** depicts the success rate and loss curve among Implicit baseline(NONE: blue), Explicit CoT baseline(FWD BACKTRACK: orange), and our method(BWD MARK: green), respectively. For the success rate of our method, we provide additional performance of the model checkpoints at step 125, 250, 375, 500, 625, 1250, 2500, and 3125.

## E.2 FEW-SHOT RESULT ANALYSIS

We sample 100 examples in Textualized Gridworld environments to proceed few-shot experiments. We also sample 4 examples which are used as demonstrations for few-shot prompting. Table 8, 9 and 10 each shows the performance of each models with 0, 1, 2 or 4-shot demonstration. We can see that all the language models struggle solving Gridworld path-planning with few-shot prompting, which needs a training phase with gradient update. Only GPT-o1 in NONE experiment shows meaningful result.

|            | 0-shot | 1-shot | 2-shot | 4-shot |
|------------|--------|--------|--------|--------|
| Llama 3 8B | 0%     | 0%     | 0%     | 0%     |
| Llama 3 70B| 0%     | 0%     | 0%     | 0%     |
| GPT 4o     | 0%     | 0%     | 0%     | 0%     |
| GPT-o1     | 0%     | 18%    | 34%    | 38%    |

Table 8: NONE prompting result

|            | 0-shot | 1-shot | 2-shot | 4-shot |
|------------|--------|--------|--------|--------|
| Llama 3 8B | 0%     | 0%     | 0%     | 0%     |
| Llama 3 70B| 0%     | 0%     | 0%     | 0%     |
| GPT 4o     | 0%     | 0%     | 0%     | 1%     |
| GPT-o1     | 0%     | 0%     | 10%    | 0%     |

Table 9: COT prompting result

|            | 0-shot | 1-shot | 2-shot | 4-shot |
|------------|--------|--------|--------|--------|
| Llama 3 8B | 0%     | 0%     | 0%     | 0%     |
| Llama 3 70B| 0%     | 1%     | 0%     | 1%     |
| GPT 4o     | 0%     | 0%     | 0%     | 0%     |
| GPT-o1     | 0%     | 0%     | 0%     | 10%    |

Table 10: COGNITIVE MAP prompting result

## E.3 DETAILED ANALYSIS OF EXPLORATION VS. COGNITIVE PLANNING

We implemented DFS by training language models to explicitly follow the algorithm's search pattern (Algorithm 1). This provides a controlled comparison point for evaluating structured exploration against cognitive map-based planning. Table 12 shows example outputs from the DFS implementation.

As shown in Figure 9, DFS achieves a 94.5% success rate compared to 88.5% for cognitive maps. This higher success rate likely stems from DFS's exhaustive exploration strategy. On the other hand, Figure 10 demonstrates the stark difference in planning efficiency. The cognitive map approach shows a near-linear relationship between optimal path length and actual steps taken, while DFS exhibits quadratic scaling.

| Cognitive map(best) | Exploration-based |
|:---:|:---:|
| 0.885 | **0.945** |

Table 11: Comparsion of reachability with exploration-based planning(DFS) and our best(BWD MARKING deadend cognitive map)

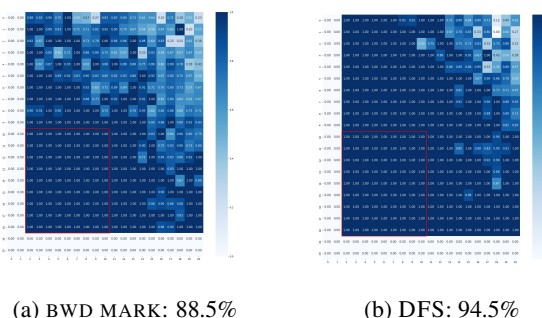

(a) BWD MARK: 88.5%  (b) DFS: 94.5%

Figure 9: Reachability plan performance between ours(BWD MARKING deadend) and DFS

| DFS Prompt | gpt: up |
|---|---|
| | human: Current:\n(0, 1)\nPossible:\n(0, 2)\nup\n(0, 0)\ndown |
| | gpt: up |
| | human: Current:\n(0, 2)\nPossible:\n(0, 1)\ndown\n(1, 2)\nright |
| | gpt: right |
| | human: Current:\n(1, 2)\nPossible:\n(0, 2)\nleft |
| | gpt: left |
| | human: Current:\n(0, 2)\nPossible:\n(0, 1)\ndown\n(1, 2)\nright |
| | gpt: down |
| | human: Current:\n(0, 1)\nPossible:\n(0, 2)\nup\n(0, 0)\ndown |
| | gpt: down |
| | human: Current:\n(0, 0)\nPossible:\n(0, 1)\nup\n(1, 0)\nright |
| | gpt: right |
| | human: Current:\n(1, 0)\nPossible:\n(0, 0)\nleft\n(2, 0)\nright |
| | gpt: right |
| | human: Current:\n(2, 0)\nPossible:\n(2, 1)\nup\n(1, 0)\nleft |
| | gpt: up |
| | human: Current:\n(2, 1)\nPossible:\n(2, 0)\ndown\n(3, 1)\nright |
| | gpt: right |
| | human: Current:\n(3, 1)\nPossible:\n(3, 2)\nup\n(2, 1)\nleft |
| | gpt: up |

Table 12: The prompt for DFS case

## E.4 ADDITIONAL INVESTIGATIONS

**Effect of ALL inclusion** Both cognitive map constructions w.o. ALL (w.o. ALL BACKTRACK: 29.6% and 29.5% for BWD and FWD construction, respectively; w.o. ALL: 27.7% and 29.0% for BWD and FWD construction, respectively) suffer at generating the optimal plan, achieving under 30% success rate. However, every cognitive map construction with the inclusion of ALL shows better performance, with FWD MARK being the lowest among them(58.5%).

We observe a similar trend in the reachable planning test. While both baseline methods w.o. ALL performed at best 41.6% (FWD w.o. ALL BACKTRACK), the lowest performance among every cognitive

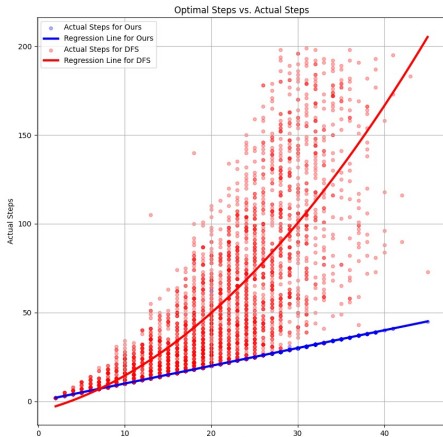

Figure 10: Number of inference steps per optimal steps for succeeded instances. We compare exploration-based planning(DFS: red) with our method(BWD MARKING deadend: blue) in the reachable planning setting. Each scattered dot denotes the succeeded instance, and the line plot denotes the regression line of the success. Note that the regression line of our method is almost identical to a $x = y$ graph, which means that cognitive maps help generating optimal performance even for the reachable planning setting.

---

**Algorithm 1** Tree search via DFS

---

$s \leftarrow start$, queue $\leftarrow [s]$
**while** $s \neq goal$ **do**
    **if** $|\text{EXPLORE}(s)| \geq 1$ **then**             $\triangleright$ If there is a sample from $s$ not visited yet
        $tmp \leftarrow \text{RANDOM}(\text{EXPLORE}(s))$
        $\text{PARENT}(tmp) \leftarrow s$
        $s \leftarrow tmp$
    **end if**
    $s \leftarrow \text{PARENT}(s)$
    queue$+ = s$
**end while**

---

map construction was 72.4% (BWD UNMARK). This implies that the **inclusion of ALL significantly enhances the planning capability, leading to more successful and efficient pathfinding.**

**Effect of BACKTRACK inclusion** Both cognitive map constructions w.o. BACKTRACK (w.o. MARKING BACKTRACK: 40.6% and 52.8% for BWD and FWD construction, respectively; w.o. BACKTRACK: 42.3% and 51.6% for BWD and FWD construction, respectively) suffer at generating the optimal plan. However, every cognitive map construction with the inclusion of BACKTRACK shows better performance, with FWD MARK being the lowest among them(58.5%).

The analysis in the reachable planning test was slightly blurry, yet there was an obvious trend. For each experiment, adding backtracking enhanced the performance of the planning in most settings(except BWD construction UNMARK). This implies that the **inclusion of BACKTRACK slightly enhances the planning capability.**

E.5   VIZUALIZATION FOR OPTIMAL PLANNING EXPERIMENTS

For optimal planning, we have only success or failure cases. Hence we only provide the success rate for each experiment.

**FWD construction**    See Figure 11 for success rate of each experiment.

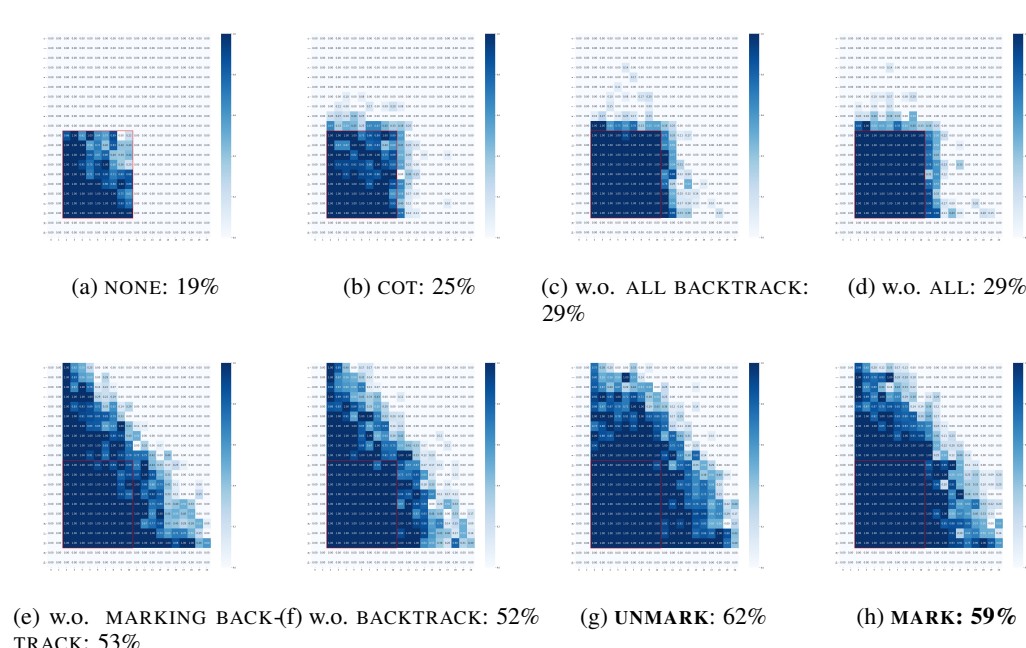

(a) NONE: 19%    (b) COT: 25%    (c) w.o. ALL BACKTRACK: 29%    (d) w.o. ALL: 29%

(e) w.o. MARKING BACK-TRACK: 53%    (f) w.o. BACKTRACK: 52%    (g) UNMARK: 62%    (h) **MARK: 59%**

Figure 11: Success rate for optimal planning, FWD construction

**BWD construction**    See Figure 12 for success rate of each experiment.

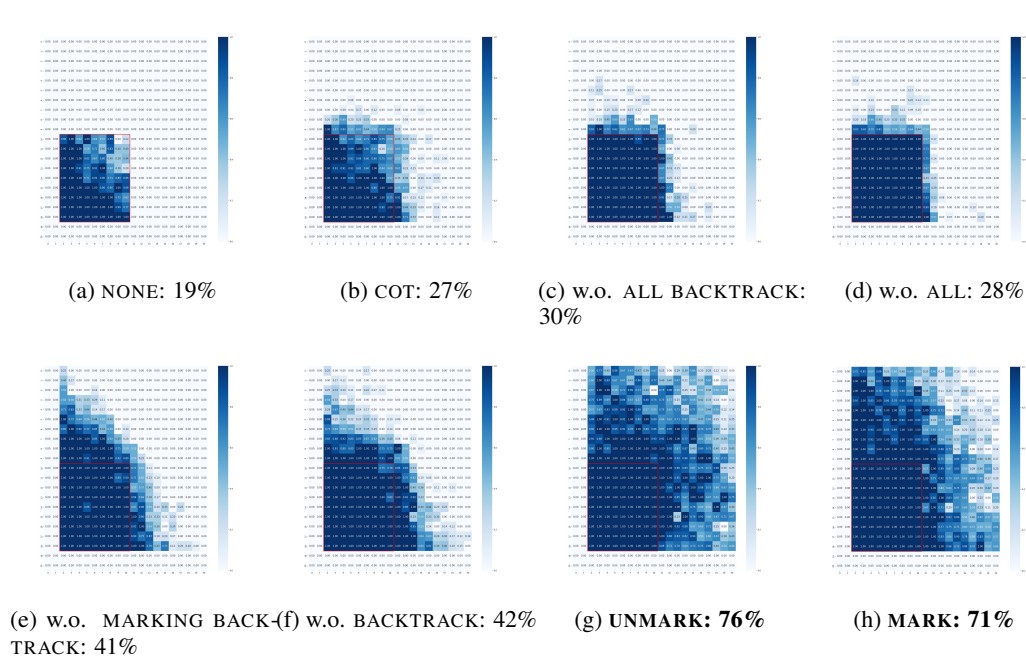

(a) NONE: 19%    (b) COT: 27%    (c) w.o. ALL BACKTRACK: 30%    (d) w.o. ALL: 28%

(e) w.o. MARKING BACK-TRACK: 41%    (f) w.o. BACKTRACK: 42%    (g) **UNMARK: 76%**    (h) **MARK: 71%**

Figure 12: Success rate for optimal planning, BWD construction

E.6   VISUALIZATION FOR REACHABLE PLANNING EXPERIMENTS

For reachable planning, we have one success cases and three different failure cases(deadend, max step, and invalid). Hence we provide the visualization of all 4 cases.

**FWD construction**   For success rate, see Figure 13. For failure cases, see Figure 14 for deadend, Figure 15 for max step, and Figrue 16 for invalid rate.

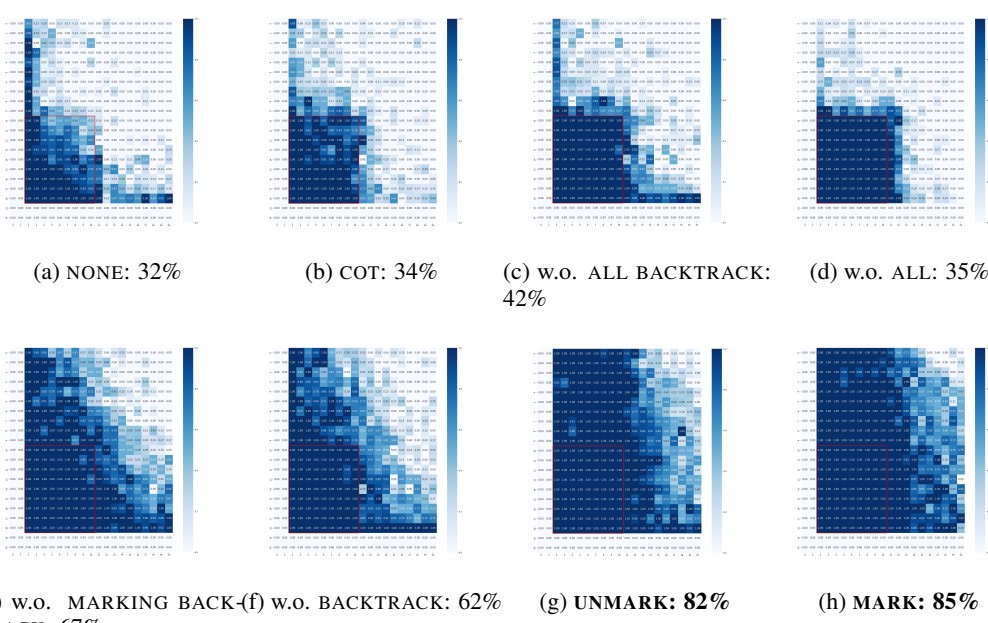

(a) NONE: 32%          (b) COT: 34%          (c) w.o. ALL BACKTRACK: 42%          (d) w.o. ALL: 35%

(e) w.o. MARKING BACK-TRACK: 67%   (f) w.o. BACKTRACK: 62%   (g) **UNMARK: 82%**   (h) **MARK: 85%**

Figure 13: Success rate for reachable planning, FWD construction

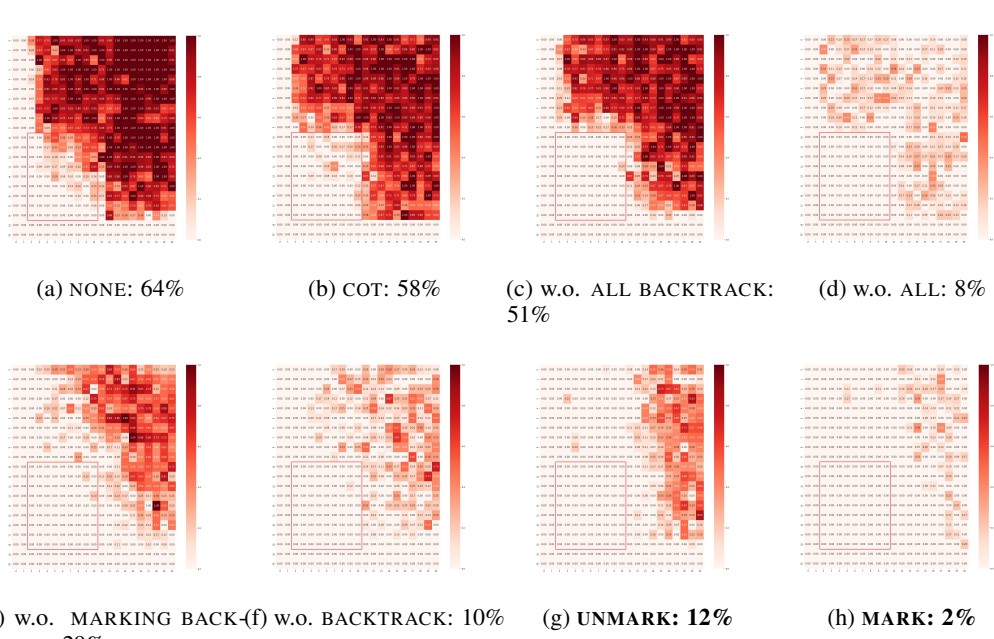

(a) NONE: 64%          (b) COT: 58%          (c) w.o. ALL BACKTRACK: 51%          (d) w.o. ALL: 8%

(e) w.o. MARKING BACK-TRACK: 29%   (f) w.o. BACKTRACK: 10%   (g) **UNMARK: 12%**   (h) **MARK: 2%**

Figure 14: Deadend rate for reachable planning, FWD construction

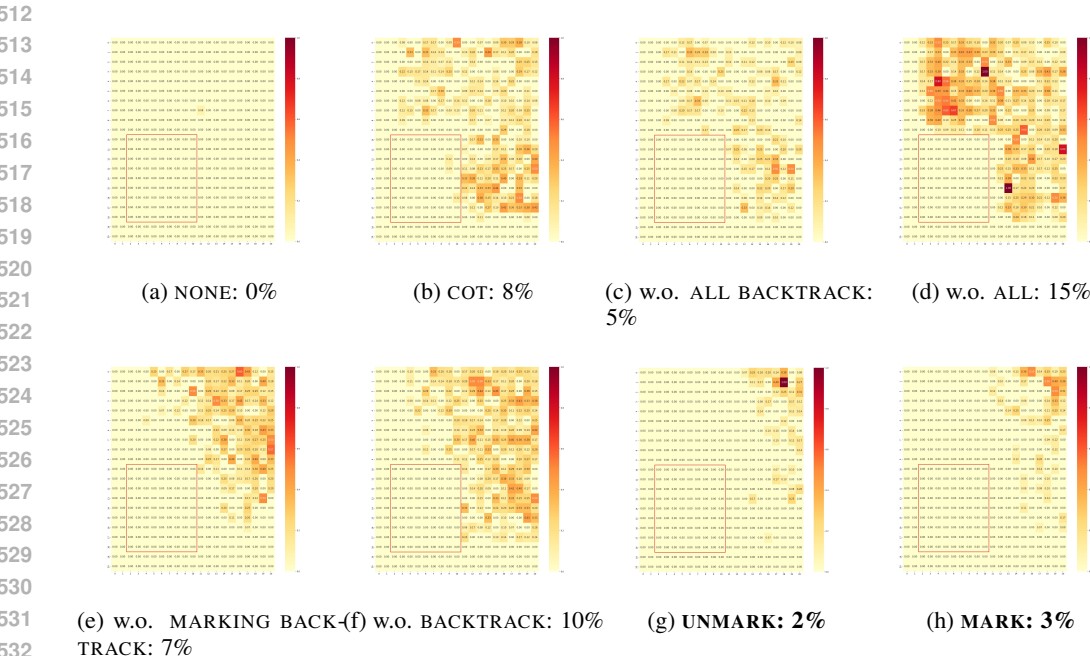

(a) NONE: 0%  (b) COT: 8%  (c) w.o. ALL BACKTRACK: 5%  (d) w.o. ALL: 15%

(e) w.o. MARKING BACK-TRACK: 7%  (f) w.o. BACKTRACK: 10%  (g) UNMARK: **2%**  (h) MARK: **3%**

Figure 15: Max step rate for reachable planning, FWD construction

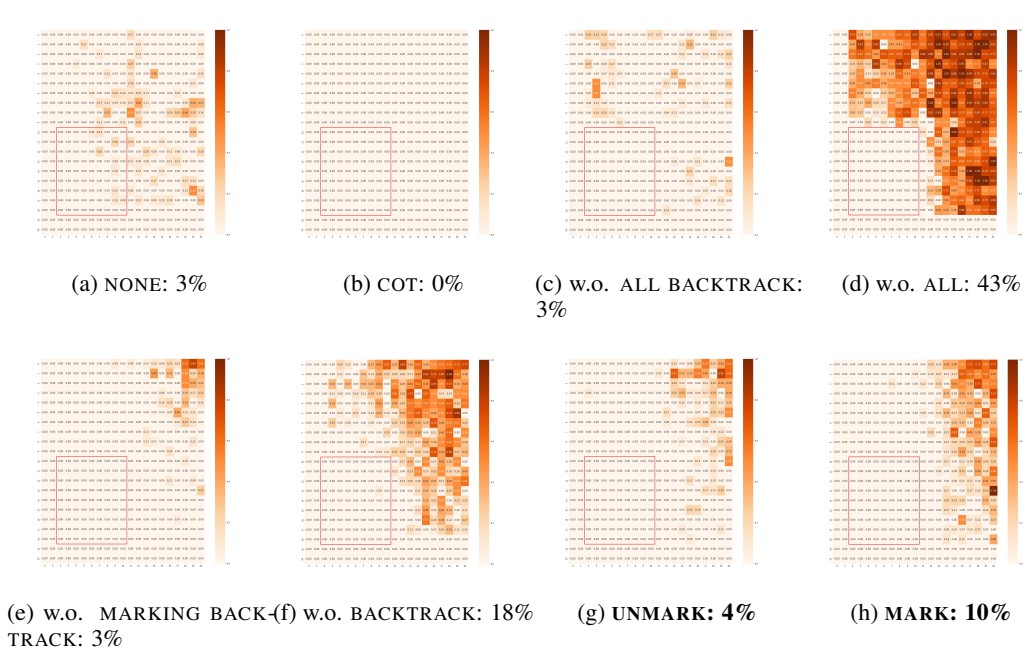

(a) NONE: 3%  (b) COT: 0%  (c) w.o. ALL BACKTRACK: 3%  (d) w.o. ALL: 43%

(e) w.o. MARKING BACK-TRACK: 3%  (f) w.o. BACKTRACK: 18%  (g) UNMARK: **4%**  (h) MARK: **10%**

Figure 16: Invalid rate for reachable planning, FWD construction

**BWD construction** For success rate, see Figure 17. For failure cases, see Figure 18 for deadend, Figure 19 for max step, and Figrue 20 for invalid rate.

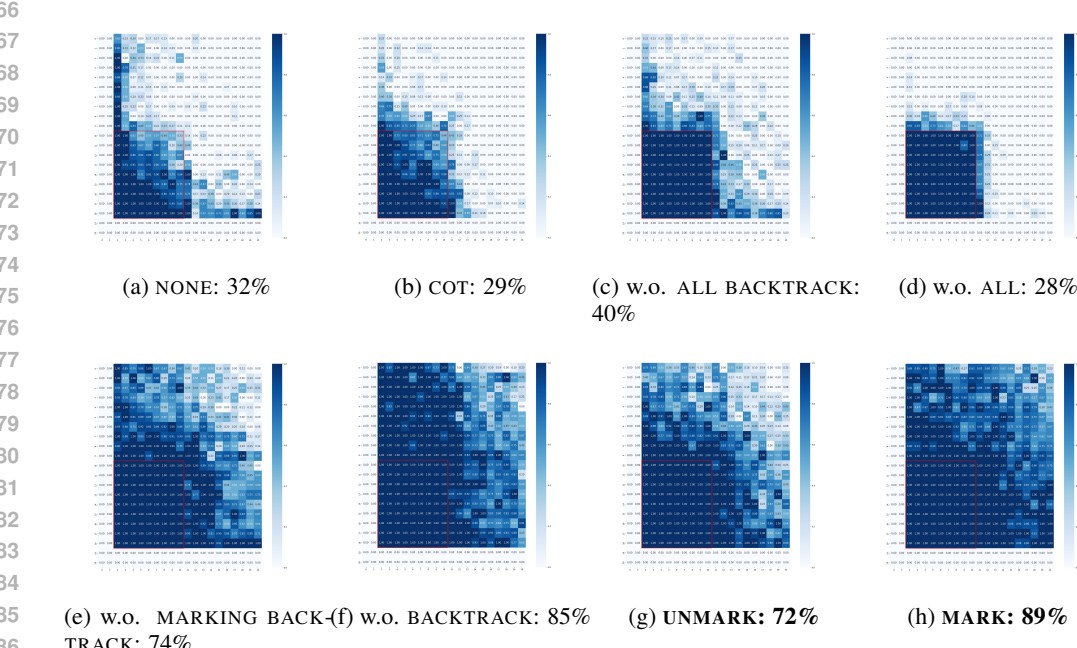

(a) NONE: 32%  (b) COT: 29%  (c) w.o. ALL BACKTRACK: 40%  (d) w.o. ALL: 28%

(e) w.o. MARKING BACK-TRACK: 74%  (f) w.o. BACKTRACK: 85%  (g) UNMARK: **72%**  (h) MARK: **89%**

Figure 17: Success rate for reachable planning, BWD construction

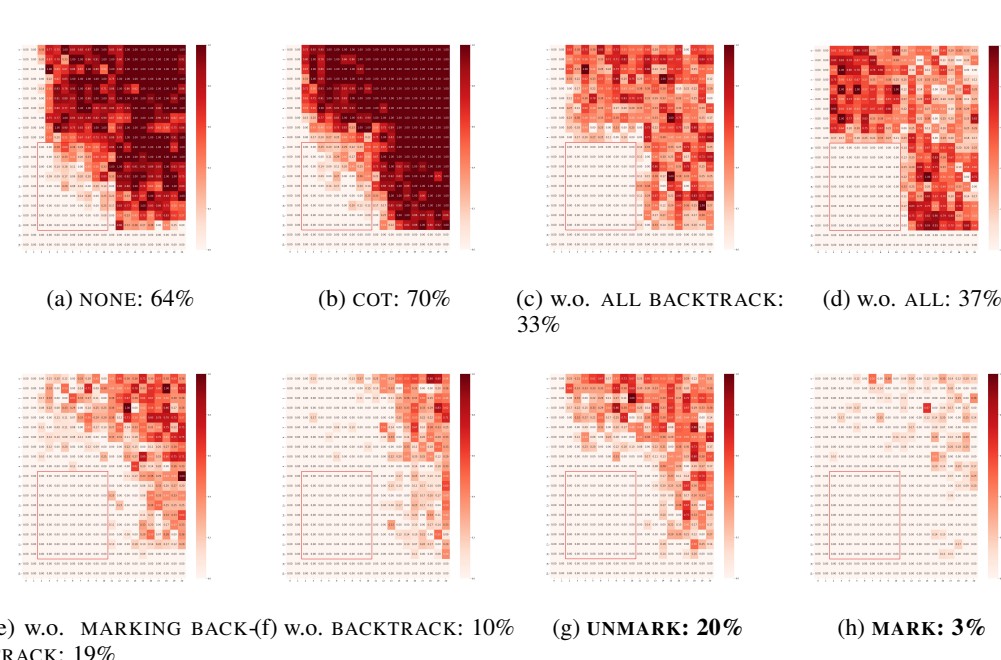

(a) NONE: 64%  (b) COT: 70%  (c) w.o. ALL BACKTRACK: 33%  (d) w.o. ALL: 37%

(e) w.o. MARKING BACK-TRACK: 19%  (f) w.o. BACKTRACK: 10%  (g) UNMARK: **20%**  (h) MARK: **3%**

Figure 18: Deadend rate for reachable planning, BWD construction

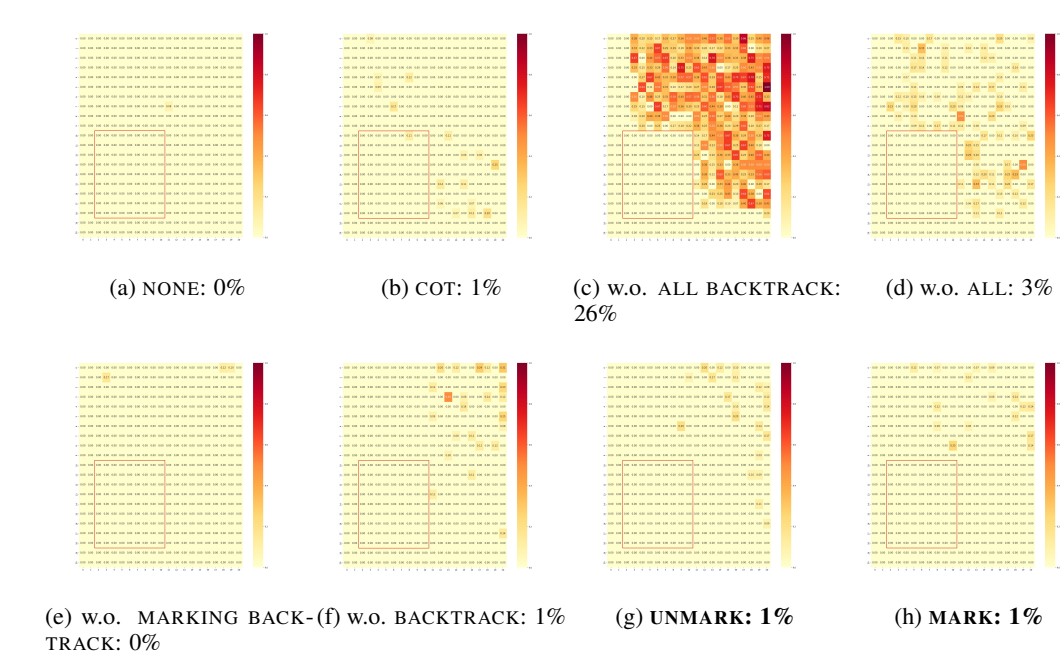

(a) NONE: 0%    (b) COT: 1%    (c) w.o. ALL BACKTRACK: 26%    (d) w.o. ALL: 3%

(e) w.o. MARKING BACK-TRACK: 0%    (f) w.o. BACKTRACK: 1%    (g) **UNMARK: 1%**    (h) **MARK: 1%**

Figure 19: Max step rate for reachable planning, BWD construction

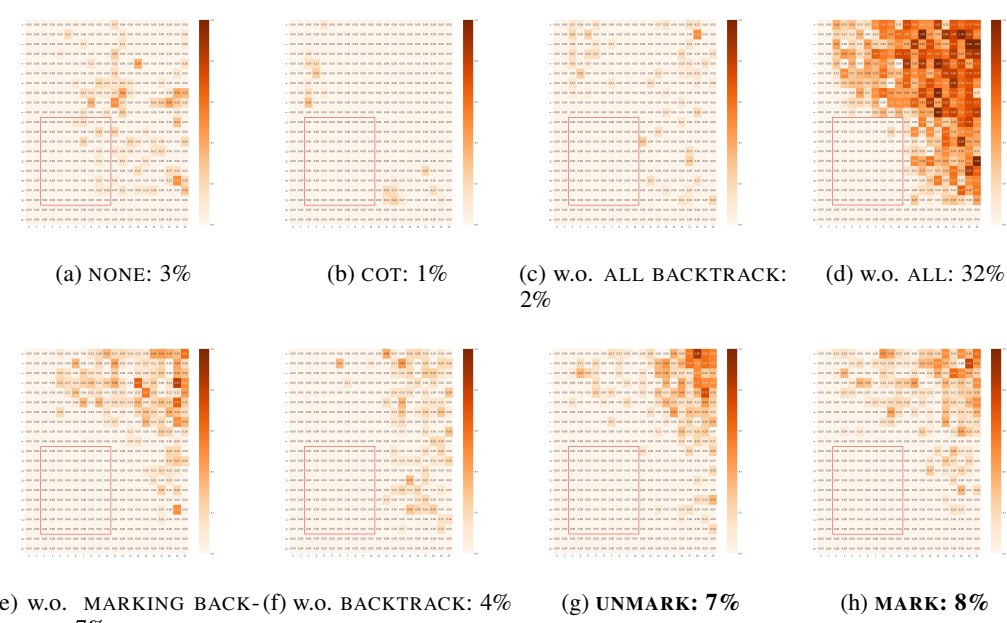

(a) NONE: 3%    (b) COT: 1%    (c) w.o. ALL BACKTRACK: 2%    (d) w.o. ALL: 32%

(e) w.o. MARKING BACK-TRACK: 7%    (f) w.o. BACKTRACK: 4%    (g) **UNMARK: 7%**    (h) **MARK: 8%**

Figure 20: Invalid rate for reachable planning, BWD construction

