# OpenReview forum: "How language models extrapolate outside the training data: A Case study in Textualized Gridworld"
_ICLR.cc/2025/Conference — Submitted to ICLR 2025_

### Official Review · Reviewer_JNpd · 2024-11-04

**Soundness:** 3
**Presentation:** 3
**Contribution:** 3
**Rating:** 8
**Confidence:** 4

**Summary:**

The paper investigates the generalisation & extrapolation of LLMs in a controlled setting and proposes a new method for better extrapolation in the controlled setting. Specifically, the authors investigate whether and how LLMs generalise in the task of path planning. To this end the paper uses a textual grid-world of varying sizes. The authors train the models on grid sizes of 10x10 and then evaluate the models on grid sizes up to 20x20. Direct Answer and CoT produce poor results barely generalising beyond 10x10. The author's method of `cognitive maps' shows strong performance and generalisation to the grid sizes up to 20x20. Interestingly zero-/ and few-shot prompting elicits very poor performance except for models such as o1 (however, still lower than the author's approach). Finally, the author's detail their experiments and results in detail.

**Strengths:**

Strengths:
1. Great analysis of extrapolation ability of LLMs.
2. Detailed experiments and results
3. Strong and impressive results on the controlled task of grid-world navigation.

**Weaknesses:**

Weaknesses:
1. Focus on one control task, it would be interesting to do additional experiments of the like done in `Physics of Language Models' https://physics.allen-zhu.com/
2. Question of how to apply to tasks beyond the specific control task?

**Questions:**

Question:
1. How would you cognitive map method generalise to other approaches?
2. Comparison to previous work: How does this work compare to approaches like: Act-Re (which has a fine-tuning mechanism), or StateAct, (which proposes a `state-tracking')?

---

> ### Author Response · Authors · 2024-11-20
>
> We appreciate your comments about task generallization and related works. We'd like to answer your questions:
>
> **1. Generalizability of Our Cognitive Map:**
> - Our findings opens several promising research directions:
>     * Developing more general representations of cognitive maps beyond spatial reasoning
>     * Creating architectures that naturally support tree-structured thinking
>     * Exploring how cognitive maps could enhance other types of complex reasoning
> - For your information, we are currently pursuing one such direction through modifying current sequential language modeling to enable native generation of decision trees:
>     * Training language models to generate sequences of actions requiring expansion, rather than single actions in traditional sequential modeling
>     * Implementing separated generation during inference so that different branches evolve independently (similar to beam search, but in a sentence level)
>     * While detailed architecture and performance analysis will be presented in future work, initial tests on challenging reasoning/planning domains (Gridworld, Game of 24, ProntoQA) show promising results
> - We also appreciate your suggestion on Physics of Language Models
>     * Our direct application could be Part 1: Learning Hierarchical Language Structures [1]
>     * We would try pretraining a GPT-2 from scratch to learn such hierarchical language structures as a future work
>
> **2. Comparsion with previous works:**
> - While the main goal of previous works such as Act-Re [2] or StateAct [3] is to interact with the environment to track and self-refine its own plan to the goal (online planning), our goal is to simulate the whole plan beforehand (offline planning)
> - Also, our main analysis is to probe the capability of the language model to “extrapolate” in complex environments in controlled setting, while previous works focus on the performance of the agent in a more practical setting
>
> References:
>
> [1] Zeyuan Allen-Zhu and Yuanzhi Li. “Physics of Language Models: Part 1, Learning Hierarchical Language Structures.” arXiv, 2024
>
> [2] Zonghan Yang and Peng Li and Ming Yan and Ji Zhang and Fei Huang and Yang Liu. “ReAct Meets ActRe: When Language Agents Enjoy Training Data Autonomy.” arXiv, 2024
>
> [3] Nikolai Rozanov and Marek Rei. “StateAct: State Tracking and Reasoning for Acting and Planning with Large Language Models.” arXiv, 2024

---

> > ### Comment · Reviewer_JNpd · 2024-11-26
> >
> > Thank you for providing a response to our questions. We will be interested in your future work as well.
> >
> > As a follow-up question to improve this work as well. It would be very interesting to compare this work also against work such as "Exploring Length Generalization in Large Language Models". (https://arxiv.org/abs/2207.04901)

---

> > > ### Author Response · Authors · 2024-12-01
> > >
> > > Thank you for this valuable suggestion and for pointing us to the work on length generalization in LLMs. We agree this would be an interesting direction for comparison and extension.
> > >
> > > To clarify our current focus: While length generalization is indeed important, our primary objective has been to explore how Chain-of-Thought prompting can help language models tackle problems beyond their inherent computational constraints. Specifically, we focus on problems outside the TC0 complexity class (which includes maze path planning in our examples).
> > >
> > > We believe that addressing length generalization for tasks within TC0 is best approached through structural modifications to the transformer architecture itself. Recent work on attention manipulation (https://arxiv.org/pdf/2310.16028) and position coupling (https://arxiv.org/abs/2405.20671v2) has shown promising results in this direction. Our work complements these structural approaches by targeting problems of higher circuit complexity while maintaining existing LLM architectures and leveraging CoT prompting. While theoretical work has established that CoT can solve these higher-complexity problems (https://arxiv.org/abs/2402.08164), the optimal configuration for doing so has remained an open question until now.
> > >
> > > We view progress in true extrapolation as requiring advances in both architectural modifications and CoT configurations. We appreciate your suggestion to incorporate length generalization analysis and will consider this as an important direction for future work.

---

### Official Review · Reviewer_4NLx · 2024-11-04

**Soundness:** 3
**Presentation:** 2
**Contribution:** 3
**Rating:** 5
**Confidence:** 3

**Summary:**

This work aims to test whether LLMs are capable of planning on out-of-distribution (extrapolated) data, and proposes a new method for fine-tuning LLMs to more effectively plan and generalize to new data. The authors propose to use a textualized gridworld as the domain of study, and OOD generalization is tested by training models on (N x N) grids and testing on (A x B) grids, where A and B may be greater than N. They test two task varients, offline and online planning, where in the first a model outputs an entire plan in one shot, and in the latter a model iteratively outputs a single action, then feedback from the environment is appended to the context, then the model produces another action, and so on. Their method consists of three stages where the LLM chains actions to and from the goal. The authors test a few variants on this method and compare to two baselines, one being a simpler CoT prompt. They fine-tune LLMs with data in their cognitive maps format, and find that with 500-1000 training steps the LLMs reach peak performance. Their method vastly exceeds the performance of their baselines in OOD generalization.

**Strengths:**

- Paper is well-motivated, and the question of whether in-context planning extrapolates to new data is compelling.
- Task seems appropriate for studying how algorithms succeed/fail to extrapolate.
- Problem formulation is thorough and described well
- A number of experiment details and different analyses provided in the Appendix.

**Weaknesses:**

- The authors make strong claims about "simulation" in reasoning, but do not sufficiently describe what this means or provide concrete justifications for their claims. E.g.
	- "The model [o1]’s success is attributed to its ability to conduct ”internal simulation” before providing an answer." I haven't seen any work making this claim, and none is cited here.
	- "These observations collectively imply that cognitive maps tap into a form of simulative reasoning that is fundamentally different from the sequential logic typically employed in CoT." I'm not sure how these results support this claim, or what "simulative reasoning" or "sequential logic" mean.
- The authors' description of OpenAI's o1 model seems off in my reading, along with the interpretation of results. To my understanding, o1 is not "a tree-searching augmented LLM capable of internal simulation before planning". The OpenAI o1 blog post [1] specifies that the model uses a (seemingly more or less traditional) CoT that is merely hidden from the user, and even gives examples of these chains. Perhaps there's something I'm missing here.
- For the authors' claim that this "emphasize[s] the need for cognitive map-regarded training", it is unclear to me what this approach would be like for real-world domains and applications.
- Missing citations, in particular [2] seems highly relevant since it tests planning and cognitive map representations in LLMs, and similarly finds that current LLMs struggle to do planning in-context. However, the models they test seem to fare much better than these authors' baselines. Another related work not cited is [3].
- The explanation of the Cognitive Map method (Section 4) wasn't very clear to me, and figures 1 & 2 didn't help me much.
	- Fig. 1 - It's not obvious to me what the takeaways are or how to map this to the main theory/methods. The only clear takeaway I see is showing extrapolation to larger grids. The grid with the shaded area (bottom-middle) seems intended to emphasize "no cognitive map / no simulative reasoning" but it looks like it could mean partial vs. full observability or something else.
	- Fig. 2 - I'm not sure whether the "cognitive map" spoken by the agent is distinct, or in a different text format, from the steps printed in the "Output" panel. The numbers mentioned in the caption aren't shown in the figure. The figure could also reference the optimal vs. reachable plan distinction mentioned in the caption.



[1] https://openai.com/index/learning-to-reason-with-llms/

[2] Momennejad et al. (2024). Evaluating cognitive maps and planning in large language models with CogEval

[3] Yamada et al. (2023). Evaluating spatial understanding of large language models.

**Questions:**

- What does "simulative reasoning" mean, and how is it different from "sequential logic" as in traditional CoT?
- What evidence is there that OpenAI-o1 does "tree search" or "internal simulation" different from traditional CoT?
- How does this work relate to [2], and what accounts for differences between the authors' results compared to [2]?
- How might the authors see their methods being used at scale and/or with real-world domains? Would "cognitive map-regarded training" be applicable to general-purpose LLMs like o1 and Claude, or would it only be used in a domain-specific fine-tuning stage?

---

> ### Author Response · Authors · 2024-11-23
>
> We greatly appreciate your thoughtful comment. We outline the key revisions made in response to your concerns. We also revised the paper upon your comments and concerns, so it would be much appreciated if you take a look.
>
> **1. Clarification on Simulation and Cognitive Maps:**
> - We agree that "simulative reasoning" needs clearer definition
> - In cognitive science, simulation refers to mental construction and manipulation of future states [1, 2]
> - Our implementation makes this concrete through:
>     * State-space exploration before action selection
>     * Explicit representation of possible future states
>     * Integration of these representations into decision-making
> - This differs from sequential CoT which:
>     * Relies on step-by-step verbal reasoning
>     * Doesn't construct global representations
>     * Makes local decisions without explicit future simulation
> - According to the o1 system card, o1 learned to refine their thinking process, try different strategies, and recognize their mistakes (https://openai.com/index/openai-o1-system-card/).
>     * We initially expressed this as a “simulative reasoning” of o1 in Section 6.2., but revised as "refine its reasoning process, explore alternative strategies, and iteratively recognize and correct its mistakes"
>     * It iteratively makes local decisions, and refine its own decision without explicit interaction
>     * We fixed the overstatement about the o1’s capability to do tree search in Section 1.
> - Our intention was to highlight that:
>     * O1's performance may suggest some form of structured reasoning as a CoT
>     * This aligns with our broader argument about the need for structured representations
> - Our work's value stands independently of o1's specific implementation (which is elusive)
>
>
> **2. Relationship to Prior Works on Evaluating Cognitive Maps in LLMs and Result Differences:**
> - Our work can be viewed as a following work since we evaluate the challenges language models face in demonstrating spatial reasoning , particulary in extrapolated environments that are unseen during training.
> - On top of them, our approach proposes a specific design of viable cognitive maps for path planning as a CoT for potential solutions to address these limitations.
> - We added a list of papers and discussions of prior works in evaluating cognitive maps in language models in Appendix A.4, including your suggestions.
>
> **3. Scalability and Real-world Applications:**
> - Probing mental representation of the spatial layout is foundational in cognitive science for studying cognitive maps [1 - 4]
>  - Like seminal cognitive science studies, our work provides valuable insights about a specific cognitive capability in a controlled environment
>  - The scientific method often progresses from controlled experiments to broader applications - many breakthrough cognitive science papers focused solely on Gridworld experiments
> - The value lies in definitively proving that current LLMs lack a crucial cognitive capability, which would be harder to demonstrate in more complex environments
>
> - Our paper suggests that we should **first establish what cognitive capabilities are missing (through controlled experiments)**, then develop scalable architectures to enable them
> - We agree that both types of generalization are important, but our work specifically addresses a fundamental limitation in current language models' cognitive abilities
> - We revised the overall storyline of the paper to better deliver our main points, throughout Abstract, Section 1, and Section 7
>
>
> **4. Future Work and Scalability:**
> - Our findings highlight a clear path forward: language models need architectural innovations to support cognitive map-like structures
> - This insight opens several promising research directions:
>     * Developing more general representations of cognitive maps beyond spatial reasoning
>     * Creating architectures that naturally support tree-structured thinking
>     * Exploring how cognitive maps could enhance other types of complex reasoning
> - We revised our detailed discussion of future work throughout Section 6, and summarize them in Section 7
> - For your information, we are currently pursuing one such direction through modifying current sequential language modeling to enable native generation of decision trees:
>     * Training language models to generate sequences of actions requiring expansion, rather than single actions in traditional sequential modeling
>     * Implementing separated generation during inference so that different branches evolve independently (similar to beam search, but in a sentence level)
>     * While detailed architecture and performance analysis will be presented in future work, initial tests on challenging reasoning/planning domains (Gridworld, Game of 24, GSM8K) show promising results
> - These early results suggest **the cognitive map insights from our controlled study can indeed generalize to broader reasoning tasks, opening a new avenue toward cognitive language models**

---

> > ### Author Response · Authors · 2024-11-23
> >
> > (continued from the previous comment)
> >
> > **5. Figure revision:**
> > - Sorry for the confusion. We revised the overall figure to better deliver our main points without misunderstanding
> >     * We revised Figure 1 to delliver that conventional finetuning with optional CoT fails language models to extrapolate in larger environments, and our objective is to incorporate a specific CoT form that makes it happen (a.k.a. Cognitive map for path planning)
> >     * We revised Figure 2 to a) describe our design of the CoT, and b) example data instance that the model will use for training
> >     * We made a new figure (Figure 3) to further describe the two planning analysis we are conducting, and to clarify that all experiments are conducted within a single agent $\pi_\theta$.
> >
> >
> > References:
> >
> > [1] Epstein RA, Patai EZ, Julian JB, Spiers HJ. “The cognitive map in humans: spatial navigation and beyond.” Nat Neurosci. 2017 Oct 26;20(11):1504-1513.
> >
> > [2] John O'Keefe & Lynn Nadel (1978) The Hippocampus as a Cognitive Map, Oxford University Press.
> >
> > [3] Kessler, F., Frankenstein, J. & Rothkopf, C.A. “Human navigation strategies and their errors result from dynamic interactions of spatial uncertainties.” Nat Commun 15, 5677 (2024).
> >
> > [4] Kadner, Florian, et al. "Finding your Way Out: Planning Strategies in Human Maze-Solving Behavior." Proceedings of the Annual Meeting of the Cognitive Science Society. Vol. 45. No. 45. 2023

---

> > > ### Author Response · Authors · 2024-11-25
> > >
> > > Also we would further like to make a note for the baseline difference between Momennejad et al. and ours.
> > >
> > > I assume you are referring to the few-shot experiment of our paper (Appendix E.2). Since that experiment was done with Optimal planning analysis, we will constrain my reply to the specific setting.
> > >
> > > Our experimental setup presents significantly higher complexity and cognitive demands compared to Momennejad et al.'s analysis in several crucial aspects:
> > >
> > > **1. Task Complexity and Success Criteria:**
> > >
> > > - Momennejad et al. only required selecting the optimal first move
> > > - Our task demands planning the complete optimal action sequence to reach the goal
> > > - For a problem with branching factor $n$ and depth $d$ (naively saying):
> > >    * Their random baseline success rate: $1/n$
> > >    * Our random baseline success rate: $1/n^d$
> > >
> > > - This exponential difference makes our task substantially more challenging, as the probability of random success approaches zero with increasing depth
> > >
> > > **2. Environment Scale:**
> > > - Their environments were limited to 21 nodes maximum
> > > - Our environments scale up to 400 nodes (20×20 grids)
> > > - This dramatic increase in scale requires significantly more sophisticated planning capabilities, and also longer planning scope.
> > > - We further provide detailed complexity analysis of different grid sizes in Appendix B.2
> > >
> > >
> > > **3. Information Structure and World Model Requirements:**
> > > - Their setup provided explicit node adjacency information
> > > - Our setup only provides basic environmental constraints (boundaries, start/goal positions, pits, walls)
> > > - Models must:
> > >    * Construct robust spatial representations
> > >    * Infer valid state transitions
> > >    * Maintain robust world model
> > >
> > > To sum up, we set substantially more sophisticated spatial reasoning and world modeling capabilities than the Momennejad et al.'s work. It was to stress test the pure extrapolability of the language models, and also to probe the robustness of our design of the cognitive maps for path planning as a CoT.

---

> ### Comment · Reviewer_4NLx · 2024-12-02
>
> Thanks for your responses and updates. The presentation of the paper has improved with the updated figures, which also helps with my understanding the basics of the methods. The points around o1 have been sufficiently clarified and reflected in the paper. I also appreciated the updates which ground this in related prior work. I appreciate the authors' clarification on "simulative reasoning", although I would have appreciated a pointer to where in the paper this definition was updated (if it has been updated).
>
> Scalability and real-world applications: The utility of a toy domain in AI and cognitive science depends on how well it reflects real world domains (ecological validity). This is important not only for applications, but also for understanding what exactly are the "cognitive capabilities" that are being studied.
>
> I think this paper would benefit from brief descriptions of *how* it could be applied to other domains. The improved methods description helps with my ability to make guesses at this, but it would help if the authors filled in some of these gaps. E.g. for "Exploring how cognitive maps could enhance other types of complex reasoning", I'm not sure how cognitive maps could be designed and used for other types of problems that aren't spatial reasoning. There is also this line "Cognitive science literature refers to this aspect of human cognition [System 2] as cognitive maps — mental representations of environments that enable flexible planning." which seems to imply that all cognition that's considered "System 2" relies on cognitive maps (minor: the System 1 vs. 2 distinction is also contentious among cognitive scientists [1]).
>
> I've updated my score from 3 to 5.
>
> [1] https://www.psychologytoday.com/us/blog/a-hovercraft-full-of-eels/202103/the-false-dilemma-system-1-vs-system-2

---

> > ### Author Response · Authors · 2024-12-02
> >
> > Dear Reviewer 4NLx,
> >
> > Thank you for your thoughtful feedback and for updating your score. We appreciate your detailed comments on the presentation improvements and clarifications around the methods.
> >
> > **1. Task generalizability beyond spatial reasoning:**
> >
> > We completely agree with your point about the importance of demonstrating how our approach could generalize beyond spatial reasoning. While our current implementation focuses on structured spatial reasoning as a proof of concept, we are actively exploring ways to extend cognitive maps to other domains of complex reasoning.
> >
> > To address this directly, we are currently pursuing a promising direction through modifications to sequential language modeling that would enable native generation of decision trees:
> >
> > - Training language models to generate multiple solution branches simultaneously within each reasoning step (for example, when solving a math equation, generating both "Let's solve this by factoring the quadratic expression first, then..." and "We can split this into two cases: when x > 0 and when x ≤ 0" in parallel) rather than generating one step at a time sequentially. This approach better mirrors how humans can hold multiple solution paths in mind while reasoning through a problem.
> > - Implementing separated generation during inference so different branches can evolve independently (similar to beam search but at the sentence level)
> > - Initial tests on challenging reasoning/planning domains (Gridworld, Game of 24, ProntoQA) show encouraging preliminary results
> >
> > While we initially omitted these details to maintain focus on our core contributions in a path planning domain, we will add a brief discussion of these future directions in the future version to better illustrate the potential for generalization.
> >
> > **2. Clarification on cognitive maps and System 2:**
> >
> > We also agree with your point about the imprecise phrasing regarding System 2 cognition and cognitive maps. Thank you for flagging the leap. We will revise this section to be more precise in our claims about the relationship between cognitive maps and different types of reasoning.
> >
> > Thank you again for helping us improve the paper's clarity and presentation. Your feedback has been invaluable in strengthening our work.

---

### Official Review · Reviewer_vAtR · 2024-11-04

**Soundness:** 3
**Presentation:** 3
**Contribution:** 3
**Rating:** 5
**Confidence:** 2

**Summary:**

This paper introduces a path planning task in a textualized Gridworld, which requires a simulation process to obtain human-like cognition.
They show that conventional approaches such as end2end generation and COT fail to generalize to larger environment. In stead, they design a  cognitive maps based approach to mimic human thinking process to enhance model planning abilities in extrapolated environments. Experiments show that their method can enable model to better generalize to larger sizes with better results.

**Strengths:**

- The task is clearly defined, the process is detailed, and the experiments show effectiveness of their proposed method;
- The motivation is clear and writing is explicit, with precise language, well-organized structure, and clear communication of ideas;
- The analysis is fairly thorough.

**Weaknesses:**

- My primary concern is that "extrapolation" or "generalizability" extends beyond simply increasing the grid size. Rather than merely expanding the grid, it would be more insightful to evaluate model performance on grids that differ in structure or complexity from the training set;
- The baselines used are relatively simple. Incorporating stronger methods, such as [1] or Tree of Thoughts for planning construction, would better support claims about the model's effectiveness. Additionally, clarifying why the proposed cognitive map approach outperforms previous planning methods would strengthen the argument;
- The test set lacks diversity, raising questions about the model's generalization capabilities and its applicability to real-world scenarios.


[1] Reasoning with Language Model is Planning with World Model

**Questions:**

- I didn't find these details in the paper: how was the training set constructed? How did you obtain the cognitive map of training samples?  What are its statistics (e.g., lengths of inputs, outputs, and plans)?
- From Figure 2, it appears that the cognitive map traverses the entire search tree. Could this cause issues with length if the grid size is large?
- In the results shown in Table 1, performance reaches 1 when the grid size is small. Does this suggest that limited diversity may be an issue, allowing the model to solve test samples simply by memorizing the training samples?

---

> ### Author Response · Authors · 2024-11-23
>
> We greatly appreciate your thoughtful feedback. Below, we outline the key revisions made in response to your concerns. We also revised the paper based on your concerns and requests, so take a look!
>
> **1. Methodology: Choice of Extrapolation Metric:**
> - Grid size provides an objective complexity metric, not just a dimensional increase
> - We quantitatively observed increased complexity:
>     * Larger grids require longer planning horizons and handling more potential paths, hence increased complexity
>     * We additionally analyzed that success probability decreases exponentially upon grid size
>     * You can check Section 2.2 and corresponding Appendix B.3 for the details
> - This controlled complexity measure allows us to definitively show extrapolation capabilities
> - **The task design ensures exactly one valid path, making success a clear indicator of true reasoning rather than chance**
>
> **2. Methodology: Comparison with Exploration-based planning methods:**
> - Our baseline choices reflect the paper's focus on decision-making rather than exploration
> - ToT, RAP, and similar methods are exploration techniques that:
>     * Separate tree generation (sampling) from tree searching (external search algorithms)
>     * Are bounded both by sampling coverage and heuristic search performance
> - Our cognitive map approach differs fundamentally by:
>     * Integrating the tree generation + searching process into the model generation
>     * Enabling direct decision-making rather than exploration
>     * Showing true extrapolation beyond training environments
> - Since exploration-based methods need “actual” interaction to reach the goal state, we could not experiment on optimal planning analysis - only reachable planning analysis is available
>
> - Even for reachable analysis, we could not compare to ToT and RAP with our main experiments
>     * They both struggled with Gridworld navigation - the pretrained models’ sampling tends to be overconfident in certain directions, preventing effective tree exploration
> - So as an alternative, we conducted additional experiments by enabling the language model to explicitly follow the DFS search in the gridworld
>     * DFS slightly overperformed in reachable analysis than our method
>     * However, when calculating the “optimality” of the generated plan, DFS required $O(n^2)$ steps, while ours only require $O(n)$ (n: optimal plan length)
>     * We interpret this result as a complementary feature of planning; While Exploration-based methods are better at adapting to revise its plan to reach to the goal (online planning), our methods are better at making an optimal plan at the first place (offline planning)
> - We provided the whole experiment description and analysis in Section 6.3 and corresponding Appendix E.3.
>
>
> **3. Justification: Scientific Value of Domain-Specific Studies:**
> - Probing mental representation of the spatial layout is foundational in cognitive science for studying cognitive maps [1 - 4]
>  - Like seminal cognitive science studies, our work provides valuable insights about a specific cognitive capability in a controlled environment
>  - The scientific method often progresses from controlled experiments to broader applications - many breakthrough cognitive science papers focused solely on Gridworld experiments
> - The value lies in definitively proving that current LLMs lack a crucial cognitive capability, which would be harder to demonstrate in more complex environments
> - We added a corresponding discussion about the justification in Section 2.2 and Appendix A.4.
>
> **4. Justification: Intended Scope:**
> - This work is deliberately focused on a **controlled environment** where we can make definitive claims about extrapolation
> - The choice of Gridworld allows us to:
>     * Precisely measure complexity through grid size
>     * Control for confounding variables that exist in more complex domains
>     * Definitively prove the failure of conventional approaches
> - **Attempting to simultaneously address scalability would have diluted these crucial findings and made it harder to draw clear conclusions**
> - This focused approach follows successful precedents in both cognitive science and AI research, where fundamental capabilities are first established in controlled settings before being scaled up
> - We revised the overall storyline of the paper to better deliver our main points, throughout Abstract, Section 1, and Section 7

---

> > ### Author Response · Authors · 2024-11-23
> >
> > (continue from the previous comment)
> >
> > **5. Addressing Technical Questions:**
> > - Training Data Construction:
> >     * We have detailed construction methodology throughout Appendix B.
> > - Cognitive Map Implementation:
> >     * Search tree exploration is bounded within maximum token limit, ensuring computational feasibility while probing extrapolability
> >     * We certify that every input + output token length (even in the test dataset) fall within maximum token length (8192)
> >     * You can check Appendix B.4 for the details.
> >
> > - Performance Analysis:
> >     * There is a clear distinction between training and test environments
> >     * Therefore the high performance on small test grids demonstrates **successful generalization within interpolation range**, rather than a memorization
> >     * It clearly contrasts with performance in extrapolation range, which is our main motivation
> >     * We have revised discussions regarding complexity measure and input/output size measure in Section 2.2 and throughout Appendix B.
> >
> >
> >
> >
> > References:
> >
> > [1] Epstein RA, Patai EZ, Julian JB, Spiers HJ. “The cognitive map in humans: spatial navigation and beyond.” Nat Neurosci. 2017 Oct 26;20(11):1504-1513.
> >
> > [2] John O'Keefe & Lynn Nadel (1978) The Hippocampus as a Cognitive Map, Oxford University Press.
> >
> > [3] Kessler, F., Frankenstein, J. & Rothkopf, C.A. “Human navigation strategies and their errors result from dynamic interactions of spatial uncertainties.” Nat Commun 15, 5677 (2024).
> >
> > [4] Kadner, Florian, et al. "Finding your Way Out: Planning Strategies in Human Maze-Solving Behavior." Proceedings of the Annual Meeting of the Cognitive Science Society. Vol. 45. No. 45. 2023.

---

> > > ### Comment · Reviewer_vAtR · 2024-11-26
> > > **Response to Author's Rebuttal**
> > >
> > > The authors' responses have addressed some of my previous concerns, and I have raised the Contribution Score to a 3.

---

> > > > ### Author Response · Authors · 2024-11-26
> > > >
> > > > Hello reviewer vAtR, thank you for recognizing our improved contribution. To help us fully address your concerns, could you please clarify what specific aspects still need strengthening to meet the acceptance threshold? Your rating score is still 5, and we are committed to making any necessary improvements to bring the paper up to ICLR standards.

---

### Official Review · Reviewer_3p1W · 2024-11-06

**Soundness:** 2
**Presentation:** 3
**Contribution:** 2
**Rating:** 3
**Confidence:** 4

**Summary:**

The paper investigates language models' extrapolation abilities in novel environments using a textualized Gridworld path-planning task. By introducing cognitive maps inspired by dual-process theory, the study demonstrates enhanced planning capabilities in language models over traditional Chain of Thought (CoT) approaches, particularly in extrapolated, larger environments.

**Strengths:**

1. Innovative approach using cognitive maps to emulate human-like System 2 reasoning in language models.
2. Rigorous experimental design testing multiple configurations of cognitive map construction (forward, backward) and planning tasks.
3. Experimental results demonstrate significant performance improvements.

**Weaknesses:**

I feel like the experiments can only prove that "cognitive maps" is a good method for the navigation task in Gridworld, but cannot support the claim that it is helpful for generalization.

The generalization ability of LLMs comes from large-scale pretraining on diverse tasks so that the models learn the underlying rules behind them and can generalize to unseen tasks. In this sense, evidences given by **training**  (or say, overfitting) a language model on a very specific domain seems not so convincing.

Specifically, training the model with cognitive map provides extra supervision signals, so the performance increase is reasonable on such a small dataset in a restricted domain, but it is unclear whether the conclusion will still hold when scaling up.

**Questions:**

Even for a proof-of-concept paper, some experimental results somehow demonstrating the scalability of the method would be better, e.g., providing results on different tasks / environments.

---

> ### Author Response · Authors · 2024-11-20
>
> Thanks for your thoughtful comments about generalization and scalability. We'd like to clarify several key points:
>
> **1. Validation of Domain-Specific Studies Outside AI:**
> - Probing mental representation through spatial tasks like Gridworld is foundational in cognitive science [1-4]
> - Like seminal cognitive science studies, our work provides valuable insights about specific cognitive capabilities in controlled environments
> - This approach can extend beyond navigation to any structured reasoning tasks such as games
> - The scientific method often progresses from controlled experiments to broader applications
>
> **2. Challenge to Current AI Development:**
> - Despite vast training data, current LLMs fundamentally lack extrapolation capabilities
> - This suggests we need to first focus on designing models with basic cognitive capabilities before pursuing broad generalization
> - While both complexity and task generalization matter, establishing fundamental capabilities in controlled settings is our priority
>
>
>
> **3. Controversy of the Role of the Cognitive Map and its Scalability:**
> - We agree that the role of the current version of the cognitive map is just giving extra supervision to the model
>  - Ironically, this “mere extra supervision” suddenly enables extrapolability beyond the training boundary - suggesting that the extra supervision fundamentally changes how the model represents and reasons about the environment
> - On the other hand, conventional intermediate reasoning structures alone are insufficient for extrapolation - even state-of-the-art LLMs with conventional CoT fail to extrapolate in our controlled environment
> - This implicitly tells you that current language models don’t have cognitive maps, or at least there is no conventional way to give model extra supervision that enables extrapolability, even with a vast pre-training corpus, large parameters, and enormous training compute
> - Our work is a proof-of-concept of how we can inject such extra supervision into the model
>
> **4. Intended Scope:**
> - Our controlled environment enables definitive claims about extrapolation
> - Gridworld allows us to:
>     * Precisely measure complexity through grid size
>     * Control for confounding variables
>     * Definitively prove conventional approaches' failure
> - **Attempting to simultaneously address generalizability would dilute these crucial findings**
> - This follows successful precedents in cognitive science, where fundamental capabilities are first established in controlled settings before being scaled up
>
> **5. Future Work and Generalizability:**
> - Our findings highlight a clear path forward: language models need architectural innovations to support cognitive map-like structures
> - This insight opens several promising research directions:
>     * Developing more general representations of cognitive maps beyond spatial reasoning
>     * Creating architectures that naturally support tree-structured thinking
> - For your information, we are currently pursuing one such direction through modifying current sequential language modeling to enable native generation of decision trees:
>     * Training language models to generate sequences of actions requiring expansion, rather than single actions in traditional sequential modeling
>     * Implementing separated generation during inference so that different branches evolve independently (similar to beam search, but in a sentence level)
>     * While detailed architecture and performance analysis will be presented in future work, initial tests on challenging reasoning/planning domains (Gridworld, Game of 24, ProntoQA) show promising results
> - These early results suggest **the cognitive map insights from our controlled study can indeed generalize to broader reasoning tasks, opening a new avenue toward cognitive language models**
>
> References:
>
> [1] Epstein RA, Patai EZ, Julian JB, Spiers HJ. “The cognitive map in humans: spatial navigation and beyond.” Nat Neurosci. 2017 Oct 26;20(11):1504-1513.
>
> [2] John O'Keefe & Lynn Nadel (1978) The Hippocampus as a Cognitive Map, Oxford University Press.
>
> [3] Kessler, F., Frankenstein, J. & Rothkopf, C.A. “Human navigation strategies and their errors result from dynamic interactions of spatial uncertainties.” Nat Commun 15, 5677 (2024).
>
> [4] Kadner, Florian, et al. "Finding your Way Out: Planning Strategies in Human Maze-Solving Behavior." Proceedings of the Annual Meeting of the Cognitive Science Society. Vol. 45. No. 45. 2023.

---

> ### Author Response · Authors · 2024-11-24
>
> Hello reviewer 3p1W, it has been a while since we last responded to your review. We further revised the paper based on your concerns and requests, and it would be much appreciated if you could take a look and respond if there are any other concerns left.
>
> We are writing this comment to emphasize once more that we respectfully disagree with your assessment and would like to clarify several key points (in a more concise form), and hope you read the comment and reply:
>
> **1. Distinction between Extrapolation and Generalization**
> - Our work specifically focuses on extrapolation (capability to solve problems of higher complexity than seen during training) rather than general domain generalization
> - As clearly defined in Section 1.2, extrapolation requires: (1) learning from simple demonstrations and (2) applying this knowledge to more complex environments
> - This is fundamentally different from the type of generalization achieved through large-scale pretraining, which typically enables interpolation within similar complexity levels
>
> **2. Experimental Design and Results**
> - Our experimental setup deliberately controls for environment complexity to isolate and test extrapolation capabilities
> - The significant performance gap between cognitive maps and baselines (both implicit and conventional CoT) under identical training conditions demonstrates that the improvement cannot be attributed to mere additional supervision, but the change of the mental representation of how language models plan in Gridworld environments
> - If this were simply an effect of overfitting or additional supervision, we would expect similar performance improvements in the baseline approaches, which was not observed
>
> **3. Scope and Contribution**
> - Our work presents a novel representation learning approach for spatial reasoning in language models, aligning with ICLR's focus on representation learning
> - Our work is **the first work** to propose specific CoT configurations to make conventional language models "extrapolate" in a certain domian, not just in theory
> - While we agree that generalization through large-scale pretraining is important, it represents a different research direction from our focus on structured representations for extrapolation
> - The limitations and scaling considerations are thoroughly discussed in Section 7 of our revised version

---

> ### Author Response · Authors · 2024-12-03
>
> Dear Reviewer 3p1W, the discussion period ends in 8 hours. Could you reply to our comments and revisions? We further revised the paper based on your concerns and requests, and it would be much appreciated if you could take a look and respond.

---

### Official Review · Reviewer_5Pyx · 2024-11-12

**Soundness:** 3
**Presentation:** 2
**Contribution:** 2
**Rating:** 5
**Confidence:** 4

**Summary:**

The authors study LMs ability to solve path finding  in synthetic GridWorld problems. They propose fine-tuning an LM to first produce path search traces (called cognitive map) then produce the path. These traces are obtained by running two different algorithms (called forward and backward) on these grids. They compare their method to CoT prompting for extrapolation to larger grids.

**Strengths:**

- The paper is written clearly
- The results are strong on the task: their backward version with markings of dead ends (Table-1 last column) generalizes to larger grids

**Weaknesses:**

- The **cognitive maps** have too much emphasis yet we see only a single synthetic task. How these maps related to how humans solve this task? Is there any experiment with humans to show similarity— eye gazing, asking them how they arrived at their solutions? I saw some references in the paper but did not find a strong discussion of it.
- The fine-tuning of rule based solved cognitive maps is not an ecologically valid comparison to humans, as humans are not trained with such intermediate maps but they could come up with themselves.
- The experiments start too late on Page 7, you could move most of  3.3 to appendix.
- The method is not novel as fine-tuning with intermediate structures or CoTs  do exist in the literature. The structure of CoT and how they constructed is different but specific to this task.

**Questions:**

I think this paper is well written but it over presents its relations to human cognition. I will increase my score if authors could provide a way to address these weaknesses in the revision.

---

> ### Author Response · Authors · 2024-11-19
>
> We appreciate your thoughtful feedback about our paper's relationship to human cognition. We address each point:
>
> **1. Connection to Human Cognition:**
> - While the direct connection between explicit decision tree and the mental representation of the cognitive map stays elusive, research in cognitive science shows that maze-solving tasks are particularly valuable for studying planning behavior in controlled environments [1 - 3]
> - Especially, eye-movement studies of human maze-solving reveal three key aspects of planning behavior [4]:
>     * Mental simulation through gaze patterns that reflect the maze's structure
>     * Balance between depth and breadth searches based on environmental complexity
>     * Adaptive planning strategies that adjust based on the number of alternatives
> - Our model's cognitive map implementation was designed by similar patterns:
>     * Constructs mental representations before taking actions
>     * Adapts its planning depth and breath based on environmental complexity
> - We will strengthen these connections in the paper by adding more detailed analysis of how our design aligns with human cognitive patterns
>
> **2. Ecological Validity:**
> - While true representation of human cognitive map being much more complex, the key similarity is that both humans and our model construct mental representations before taking actions, rather than using purely reactive strategies
> - Our goal is not to replicate the exact human learning processes, but to demonstrate that enabling cognitive map construction leads to better extrapolation abilities
> - Future work could explore more naturalistic ways of developing these capabilities, such as using self-training RL
>
> **3. Methodological Clarity:**
> - We agree with your suggestion about reorganizing Section 3.3
> - We will move implementation details to the appendix while keeping the core experimental results in the main paper
>
> **4. Novelty and Contribution:**
> - Our work fundamentally differs from existing intermediate structure approaches:
>     * We demonstrate that conventional intermediate reasoning structures alone are insufficient for extrapolation - even state-of-the-art LLMs with conventional CoT fail to extrapolate in our controlled environment
>     * Our cognitive map implementation is not merely a different structure, but enables a crucial cognitive capability (extrapolation) that conventional approaches fundamentally lack
>     * Furthermore, while conventional CoT can be learned through few-shot demonstrations, our cognitive map structure cannot - suggesting it represents a fundamentally different type of reasoning
> - Our implication challenges the current trajectory of AI development:
>     * Despite vast amounts of training data and sophisticated architectures, current LLMs fundamentally lack extrapolation capabilities even in simple controlled environments
>     * This suggests we need to first focus on designing language models with basic cognitive capabilities demonstrated through controlled experiments before pursuing broad generalization
>     * We hope this validates our experiment on controlled tasks rather than considering generalizability
>
> **Final remark:**
>
> We hope we've addressed your concerns about the paper's positioning and its relationship to human cognition. We appreciate your constructive feedback, which has helped us better revise our contributions and limitations. We look forward to incorporating these improvements in the revision. If you have any other questions, we're happy to address them. Thank you in advance!
>
> **References:**
>
> [1] Epstein RA, Patai EZ, Julian JB, Spiers HJ. “The cognitive map in humans: spatial navigation and beyond.” Nat Neurosci. 2017 Oct 26;20(11):1504-1513.
>
> [2] John O'Keefe & Lynn Nadel (1978) The Hippocampus as a Cognitive Map, Oxford University Press.
>
> [3] Kessler, F., Frankenstein, J. & Rothkopf, C.A. “Human navigation strategies and their errors result from dynamic interactions of spatial uncertainties.” Nat Commun 15, 5677 (2024).
>
> [4] Kadner, Florian, et al. "Finding your Way Out: Planning Strategies in Human Maze-Solving Behavior." Proceedings of the Annual Meeting of the Cognitive Science Society. Vol. 45. No. 45. 2023.

---

> > ### Author Response · Authors · 2024-11-24
> >
> > Hello reviewer 5Pyx, it has been a while since we last responded to your review. We once more appreciate your thoughtful feedback. We have substantially revised the paper to once more address your concerns and would appreciate your review of these changes:
> >
> >
> > **1. Connection to Human Cognitive Maps**
> > - We have significantly expanded Section 6.1 to discuss parallels between our approach and human cognition, particularly citing eye-tracking studies of human maze-solving [Kadner et al., 2023]
> > - These studies show humans engage in mental simulation through gaze patterns that mirror maze structure, similar to our model's structured exploration
> > - While our implementation is necessarily simplified, it captures key aspects of human planning:
> >    * Construction of mental representations before action
> >    * Dynamic adjustment of planning depth based on environmental complexity
> >    * Structured exploration of decision spaces
> >
> > **2. Ecological Validity and Training Approach**
> > - We acknowledge your valid point about the difference between our supervised approach and human learning
> > - However, our focus is not on replicating human learning processes, but rather on enabling similar computational capabilities
> > - The revised Section 6.1 clarifies that our cognitive maps demonstrate two key characteristics of human cognition:
> >    * Structured mental representation for planning
> >    * Rapid adaptation during training
> > - This aligns with cognitive science findings about human problem-solving strategies, even if the learning mechanism differs
> >
> >
> > **3. Methodological Novelty**
> > - We have clarified our contribution: while CoT exists, our work shows that specific structures of intermediate reasoning are crucial for extrapolation
> > - The revised paper demonstrates that conventional CoT approaches fail to enable extrapolation, even with similar amounts of supervision
> > - This suggests the structure of the cognitive map, not merely the presence of intermediate steps, is key to enabling extrapolation
> >
> >
> > **4. Paper Structure**
> > - Following your suggestion, we have moved technical details from Section 3.3 to Appendix C, improving the paper's flow
> > - Additionally, we have:
> >    * Moved Related Works to Appendix A
> >    * Restructured the conclusion in Section 7 for better organization
> >    * Organized the Introduction into Sections 1.1-1.3 to better explain our paper's scope

---

> ### Author Response · Authors · 2024-12-03
>
> Dear Reviewer 5Pyx, the discussion period ends in 8 hours. Could you reply to our comments and revisions? We further revised the paper based on your concerns and requests, and it would be much appreciated if you could take a look and respond.

---

### Author Response · Authors · 2024-12-01
**Friendly reminder to all reviewers who are not answering**

Hello reviewers, this is a friendly reminder that **the discussion period is ending soon**, and we once more ask you to reply to our comments and revisions. We further revised the paper based on each of your concerns and requests, and it would be much appreciated if you could take a look and respond.

---

> ### Author Response · Authors · 2024-12-01
> **General comment to all reviewers**
>
> Our work advances the field of representation learning by introducing a novel approach to spatial reasoning in language models through targeted Chain-of-Thought configurations. Notably, we demonstrate that through careful fine-tuning alone, language models can achieve extrapolation capabilities in spatial reasoning tasks - a significant finding that bridges theoretical possibilities with practical implementation. This result has important implications for representation learning and opens new research directions in the field. **Given ICLR's focus on representation learning advances, we believe our contribution aligns well with the conference's scope and could spark valuable discussions in the community.** We would deeply appreciate the reviewers' careful consideration of our revisions and responses to their concerns.

---

> ### Comment · Reviewer_JNpd · 2024-12-01
> **Comment after Reviews**
>
> Overall, we believe this work meaningful, as is reflected in our score (8). We believe some more comparison with length generalisation work would be good to include in this paper.

---

> > ### Author Response · Authors · 2024-12-02
> >
> > Thank you for the suggestion! The length of the problem corresponds to the size of the grid, so we can deduce that cognitive maps for path planning show length generalization. Although we did not directly plot the performance upon length, we will add them in the later appendix.

---

### Meta-Review · Area_Chair_Crz5 · 2024-12-18

**Metareview:**

This paper studies the generalization capabilities of data-driven models trained on a toy grid-world task. The find that a CoT approach (dubbed "cognitive maps") enables better generalization to unseen environments.

On the positive side, this paper studies an interesting (albeit toy) setting, and the experiments are generally well done. On the negative side, it is debatable how generalizable the findings are to more real-world cases. Moreover, I find the "human-like" claim unconvincing, given that there were no human experiments to support this claim. (The authors have added discussion around this point during the rebuttal, but mere discussion is not enough in my opinion).

I am therefore recommending that this paper not be accepted.

**Additional Comments On Reviewer Discussion:**

The reviewer response to the author rebuttal was unfortunately sparse, with only Reviewer 4NLx providing a substantive change to their score after the rebuttal. Like reviewer 4NLx, I appreciated the updated writing which clarified some points, but it was not enough justify acceptance.

---

### Decision · Program_Chairs · 2025-01-22

Reject